# An aqueous electrolyte densified by perovskite SrTiO₃ enabling high-voltage zinc-ion batteries

Rongyu Deng[1], Zhenjiang He[1], Fulu Chu[1], Jie Lei[1], Yi Cheng[1], You Zhou[1] & Feixiang Wu [1]✉

The conventional weak acidic electrolyte for aqueous zinc-ion batteries breeds many challenges, such as undesirable side reactions, and inhomogeneous zinc dendrite growth, leading to low Coulombic efficiency, low specific capacity, and poor cycle stability. Here, an aqueous densified electrolyte, namely, a conventional aqueous electrolyte with addition of perovskite SrTiO₃ powder, is developed to achieve high-performance aqueous zinc-ion batteries. The densified electrolyte demonstrates unique properties of reducing water molecule activity, improving $Zn^{2+}$ transference number, and inducing homogeneous and preferential deposition of Zn (002). As a result, the densified electrolyte exhibits an ultra-long cycle stability over 1000 cycles in Zn/Ti half cells. In addition, the densified electrolyte enables $Zn/MnO_2$ cells with a high specific capacity of 328.2 mAh g⁻¹ at 1 A g⁻¹ after 500 cycles under an extended voltage range. This work provides a simple strategy to induce dendrite-free deposition characteristics and high performance in high-voltage aqueous zinc-ion batteries.

Rechargeable aqueous zinc-ion batteries have promising potential to meet large-scale energy storage systems due to their low cost, high safety, as well as their biocompatibility and environmental friendliness[1–3]. Zinc metal has many advantages as the anode of aqueous batteries, including the high theoretical specific capacity of 820 mAh g⁻¹ and 5855 mAh cm⁻³, an appropriate reduction potential of −0.76 V (vs. SHE), and high crustal abundance, ~300 times that of lithium[4–6]. However, aqueous zinc-ion batteries still undergo several critical challenges, such as notorious side reactions, and detrimental zinc dendrite growth[7,8]. The side reactions are mainly caused by the high reactivity of water molecules in the aqueous electrolyte, including the evolution of hydrogen and the production of passivation layer on the surface of zinc, which results in the decline of cycle stability or even the failure of batteries[9,10]. The growth of zinc dendrites is due to the continuous deposition of zinc ions in regions with higher charge density[11,12]. In light of these concerns, it is significant to reduce the side reactions at the interface

between zinc anodes and electrolytes, as well as induce the uniform deposition of zinc ions.

To solve the problems mentioned above, great efforts have been made in the aspects of surface modification and electrolyte optimization[4,13,14]. Unlike the surface of lithium metal anodes, there is no solid electrolyte interphase (SEI) on the surface of zinc metal anodes, thus surface modification can be used as a strategy to produce artificial SEI[15]. The artificial coating layer can avoid physical contact between zinc metal anodes and electrolytes, as well as helps to regulate the electrochemical behavior at the interface[16]. Among the various materials used for surface modification, carbon materials and alloy compounds can make the electron distribution on the surface of zinc more uniform, while organic polymers have a better ability to reduce physical contact[17]. Inorganic compounds, such as metal oxides and inorganic salts, are mainly used to regulate the electrochemical behaviors at the interface between zinc anodes and electrolytes[18,19]. However, in the process of long-term cycles of batteries, the artificial

[1]School of Metallurgy and Environment, Engineering Research Center of the Ministry of Education for Advanced Battery Materials, Hunan Provincial Key Laboratory of Nonferrous Value-Added Metallurgy, Central South University, Changsha 410083, PR China. ✉e-mail: feixiang.wu@csu.edu.cn

coating layers are likely to be destroyed during Zn stripping/plating[20]. Compared with surface modification, electrolyte optimization has the advantages of simple operation and less possible damage[21]. The structural design of the $Zn^{2+}$ solvation sheath is one of the most important strategies to suppress side reactions, in which solvated water molecules are replaced by more polar molecules[13]. Many organic solvents such as dimethyl sulfoxide (DMSO), methanol, ethylene glycol (EG), and dimethyl carbonate (DMC) have been reported to reorganize the solvation structure of $Zn^{2+}$ [22–24]. For example, since the Gutmann donor number of DMSO (29.8) is higher than that of $H_2O$ (18), DMSO is preferentially solvated with $Zn^{2+}$, thus inhibiting the decomposition of solvated $H_2O$[22]. In terms of the strategies to suppress zinc dendrite growth, there are two commonly used methods, one is the addition of additives with an electrostatic shielding effect and the other is to induce the growth of Zn (002) crystal surfaces[25,26]. Cations, including metal cations and organic cations, can accumulate at tips with high charge density, thereby inhibiting zinc dendrites by electrostatic shielding[27]. The induction of Zn (002) inhibits dendrite growth because the angle of the zinc flakes deposited along Zn (002) is less than 30°, whereas the angle of Zn (101) and Zn (100) is greater than 70°, leading to dendrite growth[28]. In addition, the Zn (002) surface shows superiority over the Zn (100) and (101) surfaces in corrosion resistance due to its great stability[29]. A lot of organic molecules have been reported to induce Zn (002) surface owing to the ability to reduce the surface energy, such as DMSO, sorbitol (SBT), and propylene glycol (PG)[30–32]. Mai et al.[31] demonstrated that sorbitol can be solvated and delivers the highest absorption energy on the Zn (002) plane, which was beneficial to guide the deposition of zinc ions along the Zn (002) plane and realized superior Zn plating/stripping stability. However, most of the additives would result in voltage hysteresis and sluggish $Zn^{2+}$ migration kinetics due to the increase in viscosity[26]. Considering the key scientific relationship between surface microstructure and electrochemical performance, it is necessary to ensure the reaction kinetics of batteries based on the regulation of deposition morphology.

Here, we develop an aqueous densified electrolyte with reduced water molecular activity and high cation transference number using the metal oxide $SrTiO_3$ as an additive. Since the activity of water molecules is weakened by $SrTiO_3$ oxide particles, the electrochemical window of the electrolyte is widened, and the side reactions at the interface between zinc anodes and electrolytes are inhibited. More importantly, such a densified electrolyte can induce homogeneous and preferential deposition of the Zn (002) plane. Based on the density function theory (DFT), more exposed Zn (002) planes can facilitate dendrite-free deposition and the reduction of side reactions on the surface of zinc metal anodes, as the solvated $H_2O$ are more difficult to adsorb on Zn (002) than on other planes. In addition, although the produced electrolyte has solid-like characteristics, the kinetics of the electrochemical reaction is still guaranteed because the $Zn^{2+}$ transference number is improved. Therefore, the symmetric cell exhibits good long-cycle stability accompanied by the dendrite-free deposition using the densified electrolyte, and X-ray diffraction (XRD) results show the strongest peak intensity of Zn (002) plane, with almost no by-product peaks observed. Consequently, the Zn/Ti half-cell exhibits a high Coulombic efficiency of 99.6% at the 1000th cycle because side reactions are significantly suppressed by the produced densified electrolyte. What is more, the voltage range of full batteries, when coupled with manganese cathodes, is extended from 1.0–1.8 V to 0.8–2.0 V, thus achieving a significantly enhanced specific capacity of 328.2 mAh g$^{-1}$ even after 500 cycles at 1 A g$^{-1}$.

## Results

### The properties of the aqueous densified electrolyte

Figure 1a illustrates that $SrTiO_3$ is a cubic perovskite structure, crystallizing in the cubic Pm-3m space group. $Sr^{2+}$ is bonded to twelve equivalent $O^{2-}$ atoms to form $SrO_{12}$ cuboctahedra, sharing corners with twelve equivalent $SrO_{12}$ cuboctahedra and sharing faces with six equivalent $SrO_{12}$ cuboctahedra and eight equivalent $TiO_6$ octahedra[33]. The XRD result of the $SrTiO_3$ powder used to produce the densified electrolyte shows that the powder has a highly pure crystal structure (Fig. 1b). The particle sizes of these powders are less than 5 μm, with relatively uniform particles and good dispersion, as observed by scanning electron microscope images (Fig. S1). In particular, the aqueous densified electrolyte is prepared by mixing 2 M $ZnSO_4$ solution and $SrTiO_3$ powder in a mass ratio of one-to-one, after a typical mechanical stirring (Fig. 1c). It is worth mentioning that densified electrolytes can be composed of different aqueous electrolytes and various oxides, generally referring to the electrolyte with increased density after the addition of oxides (Fig. 1d). The produced densified electrolyte is a grayish-white homogeneous dispersion liquid of $SrTiO_3$ particles, which has some solid-like or non-Newtonian fluid features (Supplementary Fig. S2). The conductivity of the densified electrolytes is slightly smaller than that of the conventional electrolyte, and gradually decreases with the increase of $SrTiO_3$ content, which is mainly because $SrTiO_3$ is an insulating material and the viscosity of the densified electrolyte is higher than that of the conventional electrolyte (Supplementary Fig. S3). To explore the effect of $SrTiO_3$ particles on electrolytes, Raman spectroscopy was used to detect changes in the state of water molecules in the densified electrolyte. In conventional electrolytes, namely 2 M $ZnSO_4$ solution, a small part of water molecules is solvated by $Zn^{2+}$, and most of the remaining free $H_2O$ molecules form a water network through intermolecular hydrogen bonding forces[34]. Compared with the conventional electrolyte, the Raman spectra of the densified electrolyte change significantly (Fig. 1e), indicating that the state of water molecules changes accordingly under the influence of $SrTiO_3$ particles. For further investigation, a detailed quantitative analysis of Raman spectra was performed. Within the range of 2800–3800 cm$^{-1}$ in Raman spectra, peaks are derived from O-H vibrations. To be precise, the peak in the high-frequency region around 3550 cm$^{-1}$ is the low energy O-H in $H_2O$, that is, the weak hydrogen bond. The peak in the low-frequency region of 3253 cm$^{-1}$ is the high-energy O-H in water, corresponding to the strong H-bond. The peak near 3416 cm$^{-1}$ is caused by the medium H-bond[35]. In the conventional electrolyte, the strong H-bond has the highest peak strength and the widest peak area of 49% (Fig. 1f), suggesting the high reactivity of the free $H_2O$ molecules. In sharp contrast, the medium H-bond has the highest peak strength, and the area of the strong hydrogen bond is reduced to 41% in the densified electrolyte (Fig. 1g), which indicates that $SrTiO_3$ particles destroy the H-bond network structure and weaken the reactivity of free water in the densified electrolyte. To further investigate the variation of the H-bond network structure with $SrTiO_3$ content, Raman spectra of electrolytes with different $SrTiO_3$ contents (0 ~ 60 wt%) and their corresponding fitting peaks were analyzed. As shown in Fig. 1h, the strong H-bond located at 3253 cm$^{-1}$ exhibits a clear downward trend as the content increases. When the content is up to 60 wt%, the proportion of strong H-bond is only 34%, which indicates that more $SrTiO_3$ particles can damage more H-bond network of the electrolyte (Supplementary Fig. S4). In the densified electrolyte, $SrTiO_3$ particles have a good ability to adsorb water molecules according to DFT calculation (Supplementary Fig. S5). The adsorption energy of $H_2O$ on various geometrical configurations demonstrates that the binding energy of $H_2O$ adsorbed on the Ti atom of $SrTiO_3$ (110) plane is up to −1.02 eV, which is significantly higher than that of other sites (Supplementary Fig. S6), indicating that this site has the best water molecular affinity (Fig. 1i). For further study of the solvated $H_2O$ molecules, molecular dynamics simulations (MSD) were used to study the solvated structure of $Zn^{2+}$. According to the calculation results, the typical solvation structure in a conventional electrolyte is that six $H_2O$ molecules are solvated by one zinc ion, namely $Zn(H_2O)_6^{2+}$ (Fig. 1j). Therefore, $SrTiO_3$ particles adsorbs water

molecules, resulting in a change in the solvation structure of $Zn^{2+}$, the most typical structure is the $SO_4^{2-}$ into the solvation shell (Fig. 1k). The calculated radial distribution functions and the corresponding integrals of Zn-O($H_2O$) and Zn-O($SO_4^{2-}$) demonstrates that the reduction of solvated water molecules in the densified electrolyte (Supplementary Fig. S7). Combining the experimental and computational results, it can be concluded that the addition of $SrTiO_3$ particles can not only change the physical properties but also change the state of both free and solvated $H_2O$ molecules in a densified electrolyte.

To deeply investigate the interaction between $SrTiO_3$ and zinc atoms at the micro-level, theoretical calculations based on DFT were carried out. The crystal planes of (100), (110), and (111) of $SrTiO_3$ were selected as research objects, and two types of adsorption sites, bridge, and top, were considered in each crystal plane. Detailed top views of the geometrical configurations of zinc atoms absorbed on the (100) plane of $SrTiO_3$ are displayed in Fig. 2a, and the corresponding adsorption energy is shown in Fig. 2d. In this plane, the adsorption energy of zinc atoms at the bridge-O, top-O, and top-Sr sites is very low, which indicates that the affinity of (100) plane of $SrTiO_3$ with zinc atoms is poor. While the (110) plane exhibits the strongest adsorption

energy to zinc atoms, among which the top oxygen site exhibits the highest adsorption energy of −3.765 eV, indicating that the (110) plane has the best affinity for zinc atoms (Fig. 2b, e). The adsorption energy of the (111) plane provides the highest adsorption energy of 2.203 eV at the bridge-Ti site, which is significantly higher than that of the (100) plane, but slightly lower than that of the (110) plane (Fig. 2c, f). The above results show that the (110) plane has the best affinity for zinc atoms. More importantly, according to the XRD pattern of $SrTiO_3$ powder, the (110) crystal face has the strongest peak, namely the most exposed crystal face under the natural condition, demonstrating that the aqueous electrolyte densified by the addition of $SrTiO_3$ provides excellent affinity for zinc atoms.

### The benefits in the aqueous densified electrolyte

To investigate the potential application of densified electrolytes in zinc-ion batteries, a series of electrochemical characterizations were carried out. The electrochemical stability window between 2 M $ZnSO_4$ and densified electrolyte was studied by linear sweep voltammetry (LSV) tests at a scan rate of $10\,\mathrm{mV\,s^{-1}}$ on coin cells using zinc metal as reference and counter electrodes, and stainless steels as working

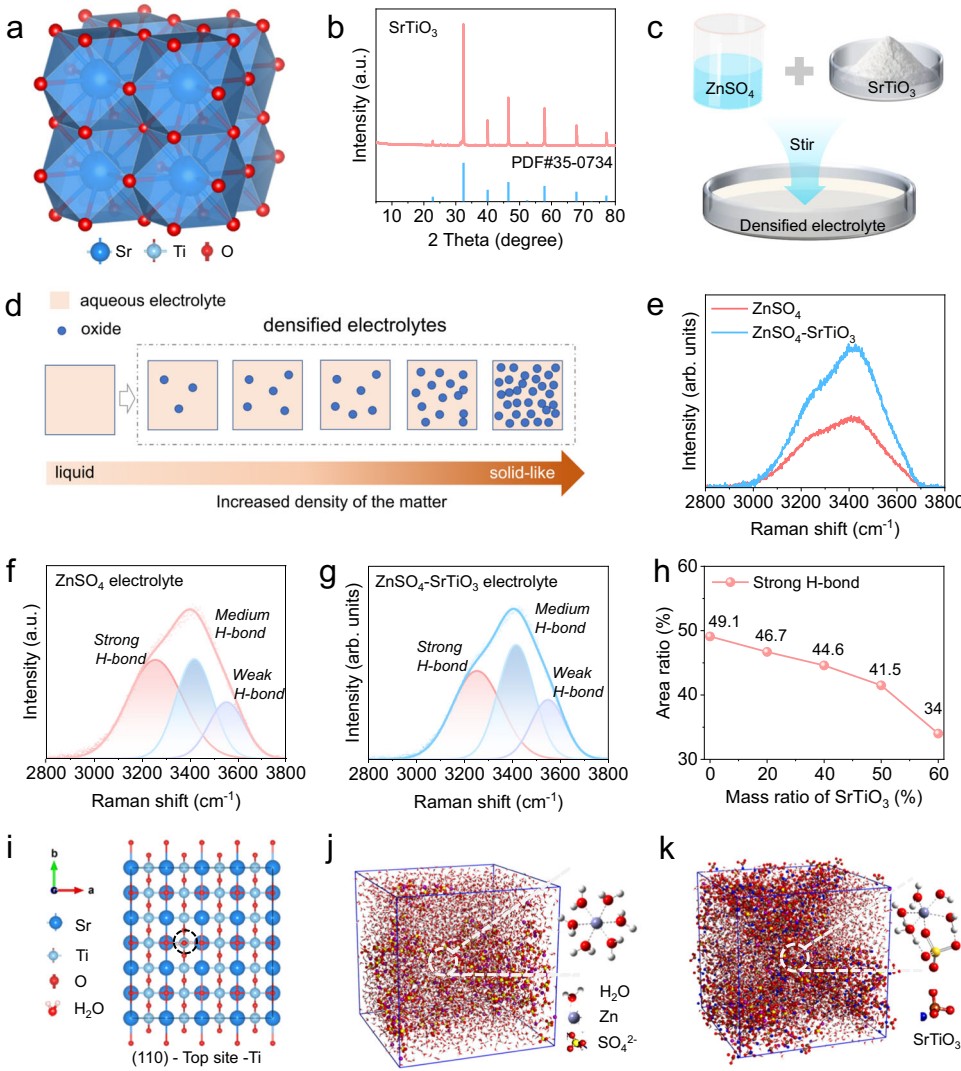

**Fig. 1 | The properties of aqueous densified electrolyte. a** The crystal structure of $SrTiO_3$. **b** The XRD spectra of the $SrTiO_3$ powder and its standard PDF card. **c** Schematic diagram of densified aqueous electrolytes. **d** Schematic illustration of densified electrolytes formed by increasing density of solution after the addition of oxide. **e** Comparison of Raman spectra of different electrolytes. Raman fit peaks of (**f**) $ZnSO_4$ electrolyte and (**g**) aqueous densified electrolyte. **h** The ratio of fitting strong H-bond area of electrolytes with various $SrTiO_3$ contents. **i** The Top view of geometrical configurations of $H_2O$ adsorbed on the Ti atom of $SrTiO_3$ (110) plane. The boxes of molecular dynamics simulations with main solvated structure of $Zn^{2+}$ in (**j**) $ZnSO_4$ electrolyte and (**k**) aqueous densified electrolyte.

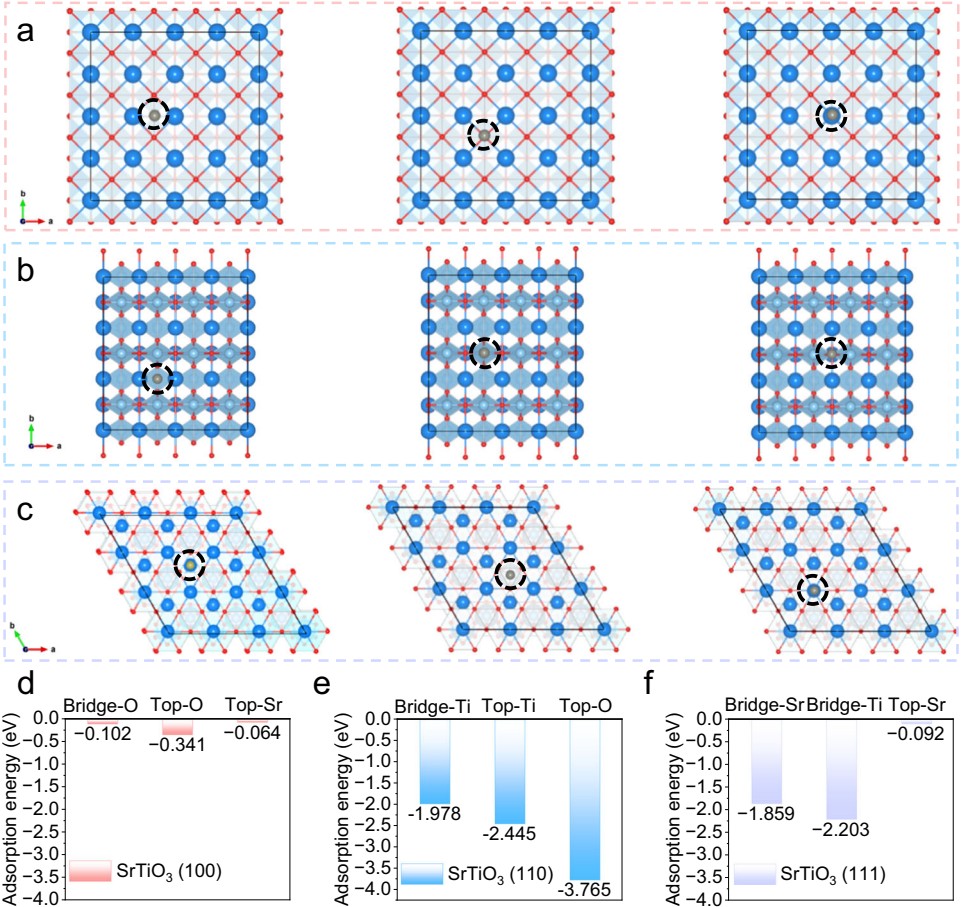

**Fig. 2 | Interaction between SrTiO$_3$ and zinc atoms at the micro-level.** Top view of geometrical configurations of zinc atoms absorbed on SrTiO$_3$ planes (the dark blue ball represents the Sr atom, the indigo ball is the Ti atom, the red ball is the O atom, and the gray ball is the Zn atom): **a** (100) plane, **b** (110) plane and **c** (111) plane. The corresponding adsorption energy between zinc atoms and various adsorption sites on SrTiO$_3$: **d** (100) plane, **e** (110) plane and **f** (111) plane.

electrodes. Within the range from −0.5 to 3.0 V, there is no obvious difference in the decomposition potential of the two electrolytes, but there is a large difference in the response current (Fig. 3a). The current for hydrogen evolution reaction (HER) in 2 M ZnSO$_4$ reaches up to 100 mA at −0.25 V, in sharp contrast, the current for the densified electrolyte is below 20 mA at −0.5 V. When the curve is magnified, the onset potential of the oxygen evolution reaction (OER) is 1.8 V for conventional electrolyte, while that of the densified electrolyte is above 2.2 V, confirming that SrTiO$_3$ particles enhance the electrochemical stability of densified electrolyte (Fig. 3b). The HER were further investigated through a three-electrode system with 30 mV reduction in the onset potential of the densified electrolyte (Fig. 3c). Side reactions between zinc metal anodes and electrolytes are the main factor of Coulombic efficiency reduction and cells failure[36]. The Tafel plots show that the densified electrolyte exhibits a lower corrosion current density, suggesting that SrTiO$_3$ can inhibit side reactions and alleviate the corrosion rate of Zn anodes (Fig. 3d). To further investigate the by-products caused by side reactions, a soaking experiment with zinc foils immersed in the electrolytes was proposed. When the Zn foil was immersed in 2 M ZnSO$_4$ for 15 days, there are visible flaky by-products on the surface (Supplementary Fig. S8a). The SEM image shows that the shape of the by-products on the surface of the immersed zinc foil are irregular polygons with average lengths of more than 100 μm. And the corresponding XRD pattern demonstrates very strong peaks of [Zn(OH)$_2$]$_3$(ZnSO$_4$)(H$_2$O)$_5$ (ZSH), even masking the peaks of Zn, suggesting that serious side reactions have occurred

on the surface (Fig. 3e). As a contrast, the zinc foil immersed in the densified electrolyte has a thin off-white layer on its surface because SrTiO$_3$ is difficult to adequately wash off (Supplementary Fig. S8b). The zinc foil in densified electrolyte shows a smooth surface without any large flake. The XRD pattern agrees well with the morphology observation and shows that there are residual SrTiO$_3$ particles and a small number of by-products on the surface (Fig. 3f). Combined Tafel plots and soaking experiments, side reactions between densified electrolytes and zinc foils significantly inhibited, which is attributed to the absorption of both solvated and free water molecules to SrTiO$_3$ particles in the densified electrolyte, according to the results of Raman spectra and MSD results.

An important but easily overlooked electrolyte parameter is the transference number. The Zn$^{2+}$ transference number is defined as the fraction of the total current carried by Zn$^{2+}$ and reflects the electromobility of Zn$^{2+}$. According to the Sand's time[37], a low Zn$^{2+}$ transference number would result in a decrease in the effective ionic conductivity, and the increased concentration polarization, further leads to the growth of zinc dendrites. The equations calculated the Zn$^{2+}$ transference number as follows[38,39]:

$$t_{\mathrm{Zn}^{2+}} = I_S(\Delta V - I_0 R_0)/I_0(\Delta V - I_S R_S) \qquad (1)$$

where $R_0$ is the resistance before polarization, $R_S$ is resistance after polarization, $\Delta V$ is the polarization potential, $I_0$ is the initial current, and $I_S$ is the stable state current. The symmetric cell using 2 M ZnSO$_4$

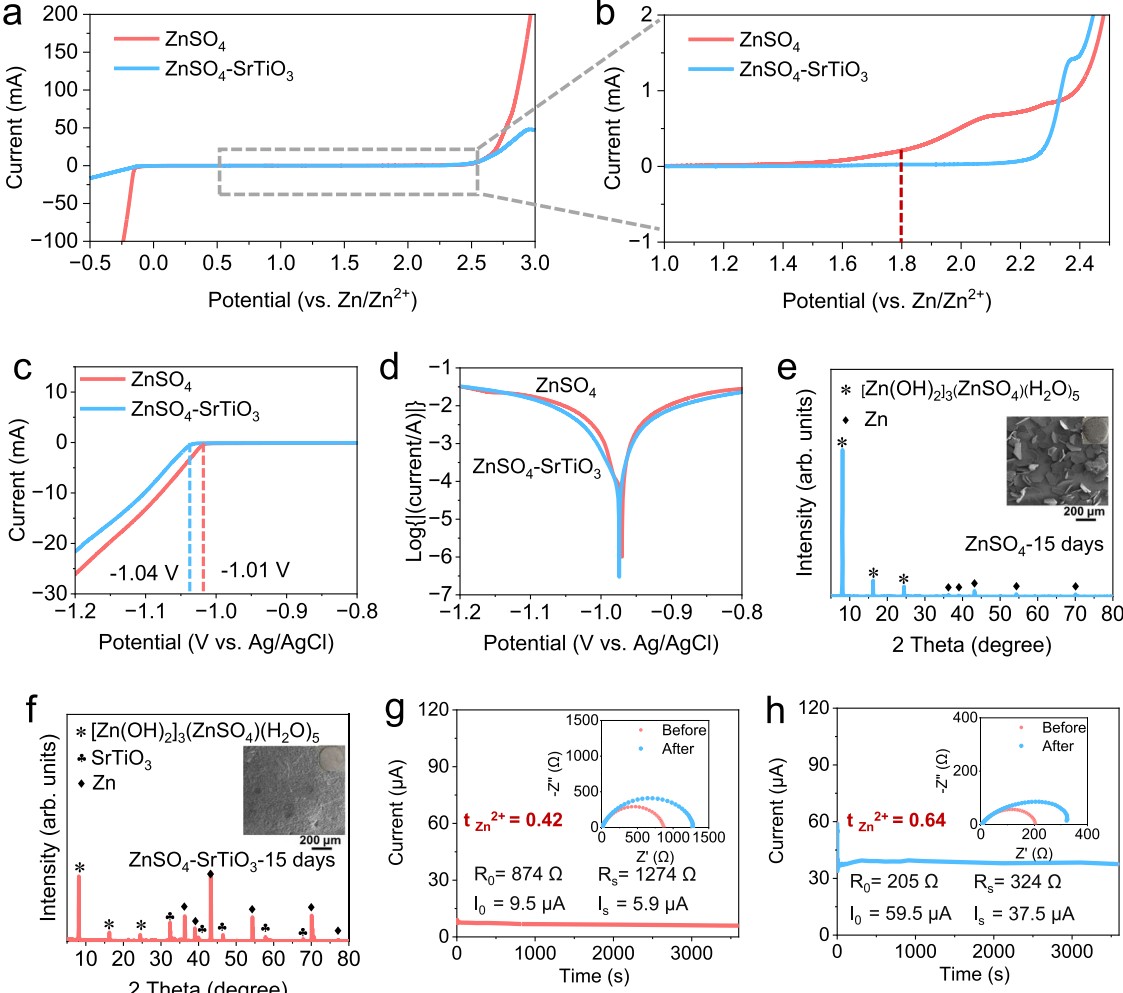

**Fig. 3 | Chemical and electrochemical properties of two electrolytes.**
**a** Electrochemical stability window. **b** Corresponding zoomed-in curves of tiny areas. **c** The linear sweep voltammetry (LSV) for test of HER. **d** Tafel curves in three-electrode cells. XRD patterns of Zn foils (the insets are the corresponding SEM images) soaked in (**e**) $ZnSO_4$ electrolyte and (**f**) the densified electrolyte. The $Zn^{2+}$ transference number measured by electrochemical polarization (the insets are associated EIS before and after the polarization of cells) at a small voltage of 10 mV in (**g**) $ZnSO_4$ electrolyte and (**h**) the densified electrolyte.

electrolyte delivers a low value of 0.42 (Fig. 3g). The transference numbers are significantly increased by adding $SrTiO_3$, and the obtained number increases to 0.64 with a significant decreased charge transfer resistance (Fig. 3h). These results indicate that the densified electrolyte has a prominent performance in broadening the electrochemical window, promoting the interface stability, and improving $Zn^{2+}$ transference number.

## The deposition behavior of zinc ions in aqueous densified electrolyte

To investigate the influences of $SrTiO_3$ particles on zinc deposition, the morphologies of deposited Zn anodes at a fixed current density of 50 µA cm⁻² in the two different electrolytes were examined by SEM. The evolution of zinc deposition in 2 M $ZnSO_4$ is shown in Fig. 4a. With the deposition time increased from 1 to 10 h, zinc dendrites grow wildly and are accompanied by a large number of by-products on the surface of Zn foils. During the first hour of zinc deposition in conventional electrolytes, although only a small amount of zinc was deposited, uneven bulk accumulation had been observed. When the deposition time was increased to 5 h, the uneven zinc deposition was aggravated, with some areas causing dendrites due to excessive deposition while the remaining areas causing vacant sites due to lack of zinc deposition. When the deposition capacity reaches

500 µAh cm⁻², in addition to the obvious dendrites, large accumulations of by-products are also observed on the surface of zinc foil. In sharp contrast, the zinc foil deposited in the densified electrolyte exhibits a flat deposition behavior, as shown in Fig. 4b. The zinc nucleation was uniform and no bulk aggregation can be observed during the first hour of zinc deposition. When the zinc deposition time reaches 5 h, a flat surface free of dendrites and by-products can be clearly observed. The deposited zinc formed a smoother and denser surface without any by-products as the deposition capacity reaches 500 µAh cm⁻². More importantly, the XRD results of zinc foils after 10 h deposition show that the preferential crystal planes of zinc deposition change significantly in two electrolytes. The XRD pattern exhibits that the ratio of the peak intensity of Zn (002) plane to Zn (100) plane ($I_{Zn(002)}/I_{Zn(100)}$) is 1.8, and there is an obvious peak of the by-product of ZSH in conventional electrolyte. The value of $I_{Zn(002)}/I_{Zn(100)}$ significantly increases to 2.9 in the densified electrolyte, suggesting that there is a large preferential growth of Zn (002) plane in the densified electrolyte (Supplementary Fig. S9). This would be attributed to the high affinity of $SrTiO_3$ for zinc atoms, leading to the effect of uniform and consistent deposition of zinc even from the early stage of nucleation.

Combined with the SEM images and XRD patterns, the deposition behavior of zinc ions in the conventional electrolyte is obviously

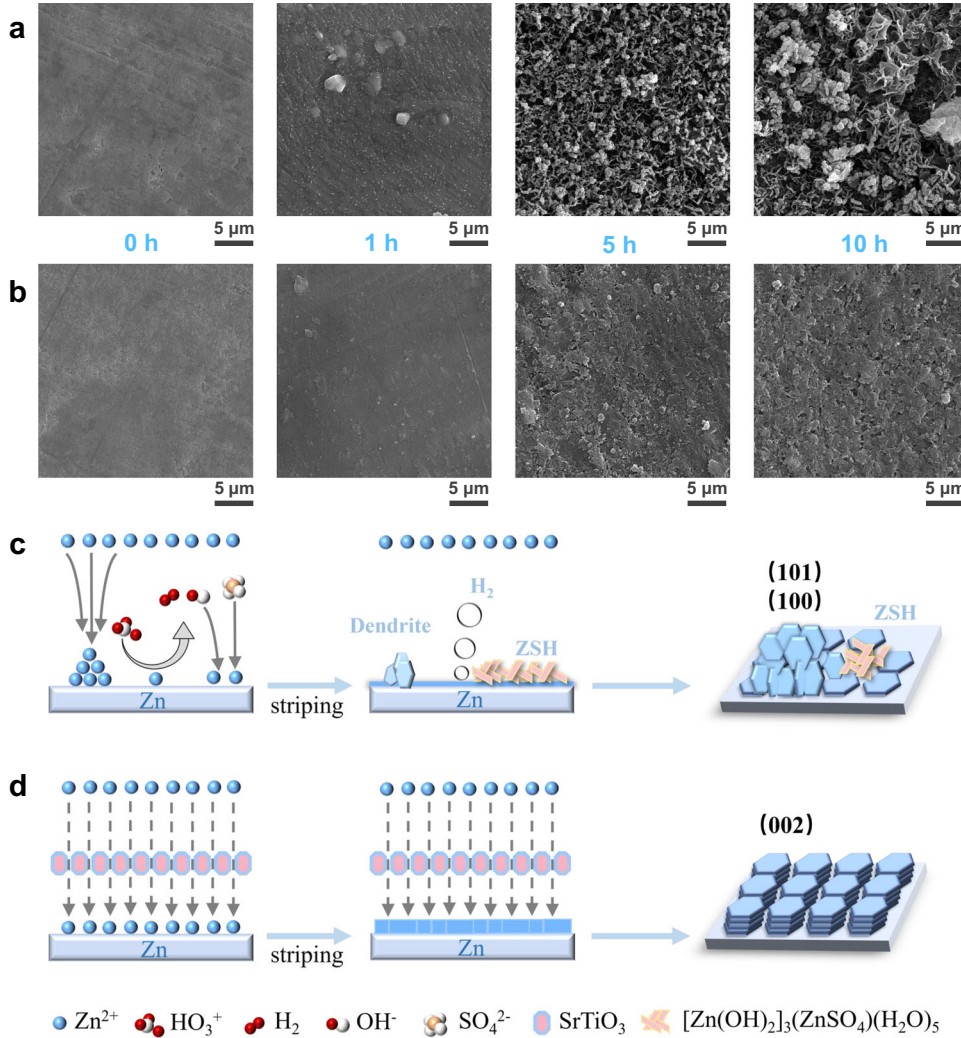

**Fig. 4 | The deposition behavior of zinc ions.** SEM images of Zn foil surface as a function of time for Zn deposition at a fixed current density of 50 µA cm⁻². **a** In the conventional electrolyte. **b** In the densified electrolyte. Schematic diagram of the zinc deposition processes in different electrolytes: **c** the conventional electrolyte and **d** the densified electrolyte.

different from that in densified electrolytes. As shown in Fig. 4c, the heterogeneous deposition of zinc ions due to the non-uniformity of the electric field and the concentration field leads to the growth of zinc dendrites in conventional electrolytes. Worse still, water molecules with high reactivity are reduced on the surface to produce hydrogen, which will not only lead to gas bloating but also cause the increase of local pH due to the production of OH⁻. Once OH⁻ ions are in contact with zinc ions and $SO_4^{2-}$, ZSH will rapidly produce and further lead to the corrosion and passivation of the Zn surface. These terrible situations take a dramatic turn in densified electrolytes, as shown in Fig. 4d. On the one hand, the side reactions between the interfaces are significantly inhibited due to the weakened activity of water molecules in the densified electrolyte. On the other hand, the $SrTiO_3$ particles have a good zinc ion affinity, which can effectively induce the deposition of zinc ions along the Zn (002) plane. When Zinc ions are deposited preferentially along Zn (002) plane, the deposited Zn flakes tend to grow at a smaller angle (-0 – 30° to the substrate), achieving uniform Zn deposition and suppression of Zn dendrites[26].

Further, DFT results demonstrate that the Zn (002) plane not only inhibits dendrite growth but also inhibits HER and reduces surface corrosion. The calculated free energy H adsorption reflects the thermo-neutral adsorption, which can imply a high activity of HER[40]. As shown in Fig. 5a and Supplementary Fig. S10, the free energy of Pt(111)

is −0.16 eV, which is very close to thermo-neutral and HER can easily occur. The free energies of Zn (101), Zn (100), and Zn (002) are 0.43, 0.72, and 1.13 eV, respectively, suggesting that Zn (002) is not conducive to H atom adsorption and thus effectively inhibits HER. However, it is worth noting that Zn deposition is easier along Zn (101) and (100) planes rather than Zn (002), because the adsorption energy of Zn atoms at (002) is significantly higher than that of the other two crystal planes (Fig. 5b), indicating that the densified electrolyte can change the adsorption behavior of zinc and expose more crystal planes of (002). In addition to inhibiting HER, Zn (002) also possessed excellent corrosion resistance. The calculation of the waste energies to strip the Zn atom from the zinc plane shows that Zn (002) requires the highest energy of 1.84 eV (Supplementary Fig. S11). The higher tripping-off energy indicates a greater internal attraction between zinc atoms. Therefore, Zn (002) plane possesses a strong chemical bond to suppress corrosion[41]. To further investigate the effect of corrosion inhibition, the adsorption energies of a typical solvation structure of $Zn(H_2O)_6^{2+}$ were calculated. Zn (002) plane has the highest energies of −1.26 eV, compared to that of the other two planes (Fig. 5c), which demonstrates that Zn (002) has great potential to reduce by-products derived from solvated water molecules.

As a result of inhibiting dendrite, HER, zinc corrosion, and by-products, the Zn/Zn symmetrical cells and Zn/Ti half cells exhibit

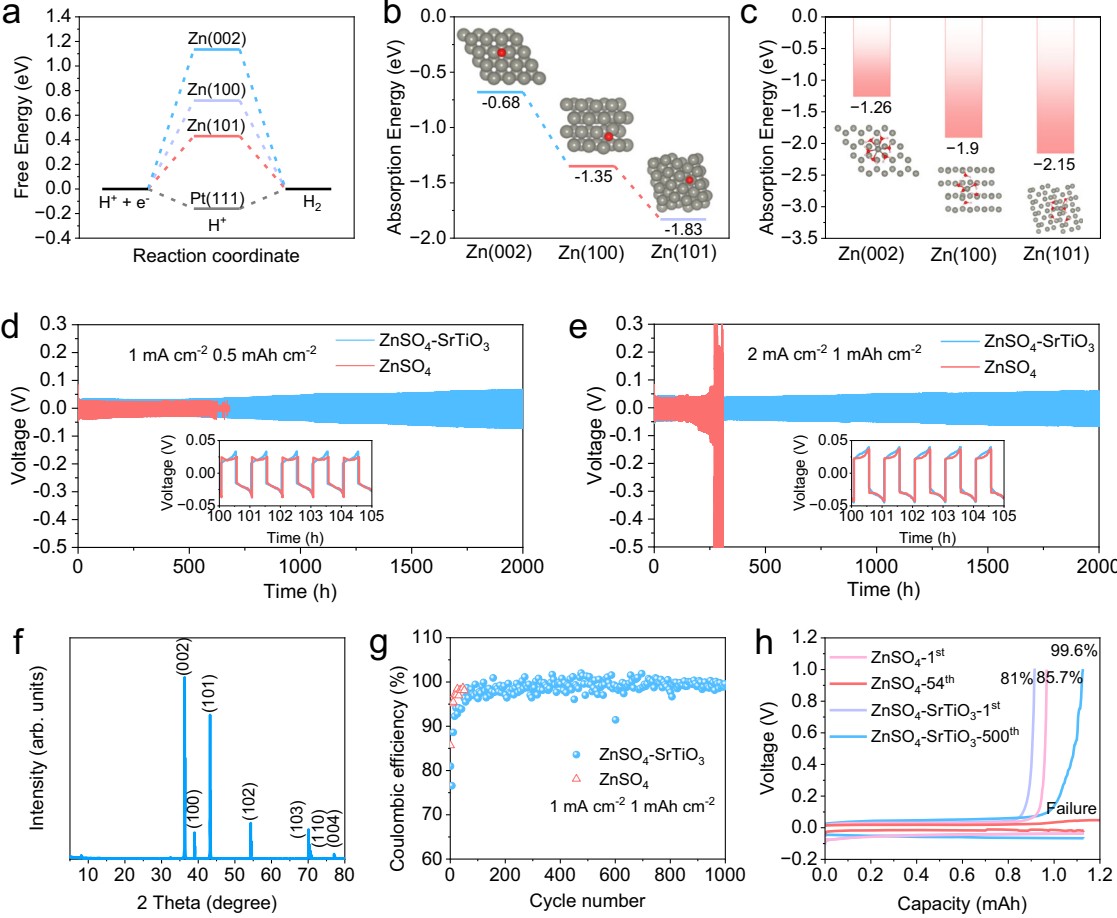

**Fig. 5 | Theoretical calculation and electrochemical performance of Zn/Zn symmetric cells. a** The free-energy of HER on Zn (101), Zn (100), Zn (002), and Pt (111). **b** The absorption energy of Zn atom at various zinc crystal planes (the gray ball represents the Zn atom in the planes, the red one is the free Zn atom). **c** The absorption energy of $Zn(H_2O)_6^{2+}$ at various zinc crystal planes (the gray ball represents the Zn atom in the planes, the other cluster is $Zn(H_2O)_6^{2+}$). Zn/Zn symmetric cells operated at different conditions: **d** 1 mA cm$^{-2}$, 0.5 mAh cm$^{-2}$ and **e** 2 mA cm$^{-2}$, 1 mAh cm$^{-2}$. **f** The XRD patterns of the zinc foil after 50 h cycle. **g** The Coulombic efficiency of Zn/Ti half cells. **h** The corresponding charge-discharge curves at different cycles.

excellent electrochemical performance in densified electrolytes. Zn/Zn symmetric configurations were carried out to study the electrochemical stability of zinc metal anodes in various electrolytes. The cell using conventional electrolytes exhibits a short lifespan of 620 h under a small galvanostatic condition of 1 mA cm$^{-2}$ and 0.5 mAh cm$^{-2}$, in contrast to more than 2000 h of the cell in densified electrolyte (Fig. 5d). However, the overpotential of the cell using the densified electrolyte is slightly larger than that of the cell using conventional electrolyte, which can be ascribed to the relatively reduced ion conductivity of the densified electrolyte. When raising the current density from 1 mA cm$^{-2}$ to 2 mAh cm$^{-2}$, the overpotential of the cell using densified electrolyte is increased appropriately, but the cycle stability is still guaranteed. The symmetric cell can still be stably cycled for more than 2000 h at 2 mA cm$^{-2}$ in the densified electrolyte, whereas the overpotential increases sharply after only 200 h in the conventional electrolyte (Fig. 5e). The drastically increasing overpotential can be attributed to the deterioration of the interface due to large accumulation of by-products produced by side reactions in the conventional electrolyte. In contrast, a slight increase in overpotential is also found in symmetric cells with densified electrolytes, this is caused by settlement of SrTiO$_3$ particles on the surface of zinc foils during a long-time cycle. The rate performance of the symmetric cell using densified electrolyte was studied from 0.1 to 10 mA cm$^{-2}$, which achieves a small overpotential of 79 mV at 5 mA cm$^{-2}$ and 139 mV at 10 mA cm$^{-2}$ (Supplementary Fig. S12). To investigate the deposition behavior of zinc

ions in the cell employing densified electrolyte, a zinc anode was obtained from the cell cycled 50 h at 1 mA cm$^{-2}$ for XRD and SEM tests. The XRD pattern demonstrates that the most exposed crystal plane is Zn (002), and no phase of any by-product can be observed, which is consistent with previous results (Fig. 5f). And the corresponding SEM images demonstrate a smooth layered stacked surface, in sharp contrast to that of zinc anode obtained from conventional electrolyte, which exhibits an uneven surface with dendrites and by-products (Supplementary Fig. S13). The Coulombic efficiencies (CEs) of Zn plating/stripping, one of the most important parameters responsible for the reversibility of electrochemical reactions, was studied by Zn/Ti half cells[42]. In the densified electrolyte, the CEs gradually increase from the first cycle, and a steady cycle of efficiency above 99% is achieved after about 100 cycles, which lasts for more than 1000 cycles (Fig. 5g). However, although the cell using conventional electrolyte exhibits a higher CE in the first cycle, the cell failed after only 54 cycles, most likely due to short circuits caused by dendrite growth. On the contrary, Fig. 5h shows that the Coulombic efficiency of the cell using densified electrolyte is 81% in the first cycle, then gradually increases to 99.6% in the 500th cycle, and the high CEs can be stably maintained for more than 1000 cycles. As the increase of the cycle number, the overpotential increases slightly, which is because of the partial settlement of SrTiO$_3$ on the electrodes due to the long-time operation. The long-term cycle stability of Zn/Zn symmetric cells at different current densities as well as the high Coulomb efficiency of Zn/Ti half-cells

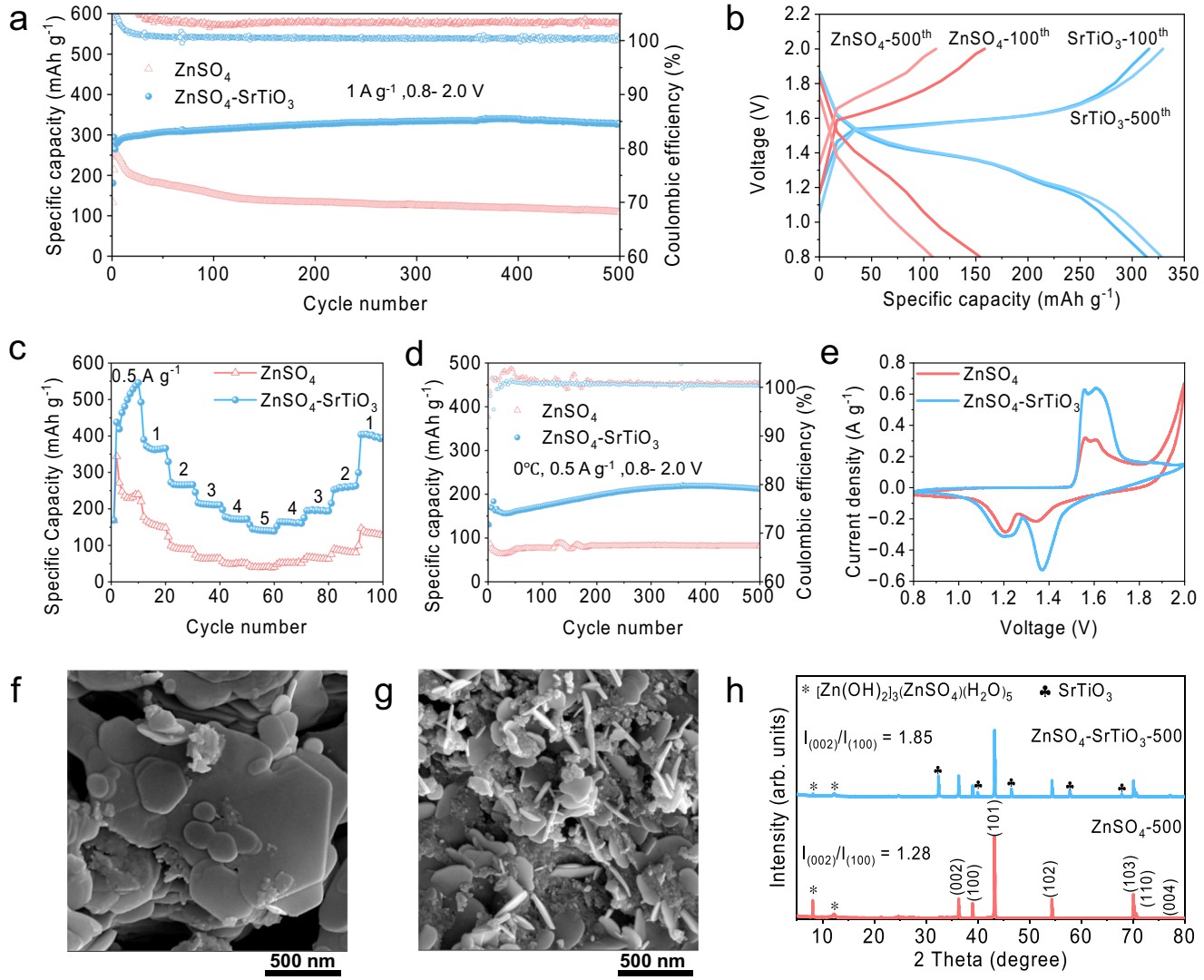

**Fig. 6 | Electrochemical performance and characterization of Zn/MnO₂ full cells. a** Long-term galvanostatic cycling performance in various electrolyte at a current density of 1 A g⁻¹. **b** Corresponding charge and discharge curves at 100th and 500th cycle. **c** The rate performance of the full cells. **d** Long-cycling performance at 0 °C. **e** The cyclic voltammetry (CV) profiles at a scan rate of 1 mV s⁻¹. The SEM images and XRD patterns of the anodes after 500 cycles in different electrolytes: **f** Densified electrolyte and **g** Conventional electrolyte, and **h** The comparison of corresponding XRD patterns.

indicate that the densified electrolyte provides considerable practical application potential in aqueous rechargeable batteries.

**Electrochemical performance of high-voltage Zn/MnO₂ full cells**
To demonstrate the practical application of the densified electrolyte, Zn/MnO₂ full cells were assembled using the conventional electrolyte (2 M ZnSO₄ and 0.1 M MnSO₄) and the densified electrolyte. According to the results of the electrochemical stability window, the densified electrolyte has a significant effect on inhibiting the electrolyte decomposition above 1.8 V. Therefore, the Zn/MnO₂ full cells are charged to a higher voltage of 2.0 V. As a result, the full cell exhibits a significant increase in specific capacity. To investigate the effect of SrTiO₃ content on the electrochemical performance, Zn/MnO₂ full cells using conventional electrolytes and with various SrTiO₃ contents were assembled. As shown in Supplementary Fig. S14, the cells using conventional electrolytes show a drastic decay, with a capacity retention rate of 43% after 500 cycles at a current density of 1 A g⁻¹. In contrast, densified electrolytes can significantly improve the specific capacity and more SrTiO₃ particles are beneficial to capacity retention. However, excessive content (60 wt%) leads to excessive densification

with visible reductions of mobility and conductivity, leading to poor electrochemical performance. The cell with 50 wt% SrTiO₃ exhibits the best cycle stability in densified electrolytes, demonstrating an initial specific capacity of 278.8 mA h⁻¹ and a slightly increased specific capacity of 328.2 mAh g⁻¹ at 500th cycle (Fig. 6a). Their corresponding charge-discharge curves indicate that there is a marked decrease of the voltage polarization in the densified electrolyte, suggesting a better redox platform and faster reaction kinetics, compared to the large polarization in conventional electrolyte (Fig. 6b). It is worth mentioning that the specific capacity of Zn/MnO₂ full cells with densified electrolyte has exceeded the theoretical value of 308 mAh g⁻¹, which is because the Mn²⁺ from MnSO₄ additive in the densified electrolyte is oxidized when the cell is charged to 2 V, thus contributing additional capacity beyond the original solid MnO₂. The redox reaction of Mn²⁺ in the densified electrolyte originates from the pH increased from 3.4 to 5.76 after the addition of SrTiO₃ (Supplementary Fig. S15), resulting in an additional electrochemical reaction as Eq. (2)[43–45]:

$$x\text{Zn}^{2+} + y\text{Mn}^{2+} + \text{H}_2\text{O} \rightleftharpoons \text{Zn}_x\text{Mn}_y\text{O} + 2\text{H}^+ + (2 - 2x - 2y)\text{e}^- \quad (2)$$

To exclude the interference of the active material $MnO_2$, the cathode containing only Super P and PVDF is designed to assemble the cell (note as SP cell). It is found that SP cells can only make additional capacity contributions when the densified electrolytes contain $MnSO_4$, $ZnSO_4$, $SrTiO_3$ and are charged to 2.0 V (Supplementary Figs. S16 and S17). The charge-discharge curves of the SP cell using conventional electrolytes have distinct plateaus around 1.99 V, which is caused by the oxidation of small amounts of $Mn^{2+}$ to $MnO_2$[46]. In contrast, the cells with densified electrolytes have a longer platform below 1.7 V, and the curve is slowly raised to 2.0 V. This is consistent with the charge-discharge curves in the whole $Zn/MnO_2$ full cells (Supplementary Fig. S18), which indicates that the electrochemical behavior of conventional electrolyte is indeed different from the densified electrolyte. The cyclic voltammetry tests of the SP cells were carried out to further study the electrochemical behavior. As shown in Supplementary Fig. S19a, after the first charge to 2.0 V, the CV curves of SP cells with conventional electrolyte are almost the same as that of $Zn/MnO_2$ full cells, indicating that $Mn^{2+}$ is oxidized to $MnO_2$. Peculiarly, a weak peak corresponding to $Zn_xMn_yO$ can be observed at the third cycle of the CV curve, which is because the pH increases due to side reactions, further indicating that the $Zn_xMn_yO$ would be produced in electrolytes with higher pH[47,48]. In contrast, the densified electrolyte demonstrates no sharp peaks at 2.0 V indicating almost no $MnO_2$ production, but distinct peaks at 1.65 and 1.35 V, which correspond to the redox reaction of $Zn_xMn_yO$ (Supplementary Fig. S19b). The XRD patterns of cathodes of SP cells after charging to 2.0 V provide further evidence (Supplementary Fig. S20). In the conventional electrolyte, the XRD pattern only shows the presence of the stainless steel but without $MnO_2$, which may be due to the amount is too small to be detected. In contrast, in addition to the peak for stainless steel and $SrTiO_3$ in the densified electrolyte, the XRD pattern exhibits several peaks, which are attributed to $Zn_xMn_yO$. Overall, these results show that the pH can be stabilized at about 5.8 after the addition of $SrTiO_3$, which leads to a continuous and reversible reaction as Eq. (2), accompanied by additional capacity from the produced solids $Zn_xMn_yO$.

To demonstrate the faster reaction kinetics, $Zn/MnO_2$ full cells were further studied at various current densities and a low temperature of 0 °C. Figure 6c compares the rate capability of the cells in various electrolytes. The densified electrolyte delivers a high specific capacity of 463 mAh g$^{-1}$ at 0.5 A g$^{-1}$ and 144.9 mAh g$^{-1}$ at a higher current density of 5 A g$^{-1}$. However, only 271.4 mAh g$^{-1}$ at 0.5 A g$^{-1}$ and 40.5 mAh g$^{-1}$ at 5 A g$^{-1}$ can be achieved in the conventional electrolyte. More importantly, when the applied current density returns to 1 A g$^{-1}$, the specific capacity of the cell using densified electrolyte recovers to 392.6 mAh g$^{-1}$. The higher specific capacity in Fig. 6c than the capacity in Fig. 6a is due to the presence of an electrochemical activation process at a relatively smaller current density, which is further proved in Supplementary Fig. S22. In contrast, the specific capacity only recovers to 129.9 mAh g$^{-1}$ in the conventional electrolyte. Besides, their corresponding charge-discharge curves demonstrate a smaller voltage polarization in the densified electrolyte (Supplementary Fig. S23). The $Zn^{2+}$ storage kinetics is further investigated by galvanostatic charging and discharging tests with a current density of 0.5 A g$^{-1}$ at a temperature of 0 °C. Conventional electrolytes deliver a low initial specific capacity of 86.5 mAh g$^{-1}$ and 81.4 mAh g$^{-1}$ at the 500th cycle, whereas a higher initial specific capacity of 130.8 mAh g$^{-1}$ and an increased value of 212.6 mAh g$^{-1}$ after 500 cycles are achieved in the densified electrolyte (Fig. 6d). The full development and continuous improvement of the batteries capacity at low temperature indicates the excellent electrochemical reaction kinetics of the densified electrolyte, which can be attributed to the significant improvement of $Zn^{2+}$ transference number. To be more practical, full cells with a high mass loading of 4 mg cm$^{-2}$ were studied in Supplementary Fig. S21. The cell using densified electrolyte delivers a specific capacity of 238 mAh g$^{-1}$ after 200 cycles at 0.5 A g$^{-1}$ with stable CEs. In contrast, the cell using the

conventional electrolyte only exhibits a low specific capacity and fails in the 170th cycle. To verify the commercial availability of the densified electrolyte, pouch cells were also assembled and studied at a constant current density of 0.5 A g$^{-1}$. The produced $Zn/MnO_2$ pouch cells possesses a higher open circuit voltage of 1.43 V than the coin cells and achieves a superb specific capacity of 367.2 mAh g$^{-1}$ after 50 cycles (Supplementary Fig. S24).

The cyclic voltammetry (CV) profiles were collected at a scan rate of 1 mV s$^{-1}$ to verify the electrochemical behaviors. $Zn/MnO_2$ full cells using both the conventional electrolyte and the densified electrolyte exhibit similar shapes with two couples of redox peaks, but the cathodic peak at 1.35 V significantly increases in densified electrolytes compared to conventional electrolytes, while the cathodic peak at 1.2 V does not change, which is due to the additional redox reaction contributing to the capacity $(xZn^{2+} + yMn^{2+} + H_2O \leftrightarrows Zn_xMn_yO + 2H^+ + (2 - 2x - 2y)e^-)$. Besides, the battery using the densified electrolyte exhibits relatively higher response current and reduction potential, indicating faster reaction kinetics, lower overpotential, and better reversibility (Fig. 6e)[49]. Subsequently, electrochemical impedance spectroscopy (EIS) demonstrates that the full cells using densified electrolytes have better reaction kinetics and electrochemical interface with smaller charge transfer impedance, compared to the cells using conventional electrolytes (Supplementary Fig. S25). Zinc metal anodes were obtained from the full cells after 500 cycles to study the electrochemical deposition behavior in the process of repeated charging and discharging. The SEM image of the anode from conventional electrolyte shows zinc deposits along Zn (100) and (101) planes due to the direction being almost vertical and accompanied by numerous scattered by-products (Fig. 6g). In contrast, there is a flat and uniform deposition morphology with a distinct hexagonal structure of Zn (002) planes on the surface of the anode from the densified electrolyte (Fig. 6f). Besides, the corresponding cross-sectional images indicate that the zinc anode using densified electrolytes demonstrates a smoother and denser surface, whereas the deposition morphology of the anode is loose and undulating in a conventional electrolyte (Supplementary Fig. S26). Their XRD patterns demonstrate that zinc is more easily deposited along Zn (002) plane in the densified electrolyte with little by-product generation, as evidenced by the increased ratio of $I_{Zn(002)}/I_{Zn(100)}$ from 1.28 to 1.85, as well as the nearly vanishing by-products (ZSH) peaks (Fig. 6h). These results reveal that the densified electrolyte provides the advantages of improving reaction kinetics, regulating directional Zn deposition, and inhibiting side reactions, thus significantly improving the electrochemical performance of aqueous zinc-ion batteries.

## Discussion

In summary, through a simple strategy of introducing $SrTiO_3$ into a conventional aqueous electrolyte, this study has developed an aqueous densified electrolyte and revealed its comprehensive effects. Since $SrTiO_3$ disrupts the H-bond network of free water and weakens the reactivity of $H_2O$ molecules in the solvated shell, the densified electrolyte inhibits the side reactions, restrains interface corrosion and passivation, and widens the electrochemical stability window. Meanwhile, the natural zincophilic property of the densified electrolyte helps to induce the uniform and preferential deposition of zinc along the Zn (002) plane, achieving a flat, dendrite-free, and densified surface morphology. Besides, the increased $Zn^{2+}$ transference number improves electrochemical reaction kinetics even at high current densities and low temperatures. As a result, Zn/Zn symmetric cells can stably work for more than 2000 h with the most exposed Zn (002) plane in the densified electrolyte at 2 mA cm$^{-2}$ and 1 mAh cm$^{-2}$. And Zn/Ti half cells using the densified electrolyte achieve remarkable cycling stability and a considerable CE of 99.6% at the 1000th cycle. Moreover, in the extended voltage range of 0.8–2.0 V, the high-voltage $Zn/MnO_2$ full cells exhibit long-term cycling stability and ultra-high specific

capacity of 328.2 mAh $g^{-1}$ at a current density of 1 A $g^{-1}$ after 500 cycles, as well as achieve an extraordinary rate capability from 0.5 to 5 A $g^{-1}$. The prepared aqueous densified electrolyte significantly improves the electrochemical performance of high-voltage zinc-ion batteries, providing a new design concept and solution for the electrolyte optimization of aqueous rechargeable batteries. More importantly, further screening on the types of oxides and their particle parameters would enhance solid-like characteristics with potentially good mechanical strength and even lead to separator-free zinc batteries.

## Methods

Preparation of aqueous densified electrolyte. 2 M $ZnSO_4$ is the conventional electrolyte, and the electrolyte for $Zn/MnO_2$ full cells should contain 0.1 M $MnSO_4$, which can inhibit the dissolution of manganese in the cathode materials. The densified electrolytes were prepared by adding 50 wt% $SrTiO_3$ to the based electrolyte, and then stirred until evenly dispersed. $ZnSO_4 \cdot 7H_2O$ and $SrTiO_3$ were purchased from Aladdin Reagent (Shanghai) Co., Ltd.

Materials synthesis. The cathode active materials were synthesized by following steps. First, 0.1 g carbon nanotubes (CNTs) were added to 60 ml of DI water for ultrasonic dispersion. Next, 0.486 g $KMnO_4$ was added and stirred for complete dissolution. Then, 1.135 g $Mn(CH_3COO)_2 \cdot 4H_2O$ dissolved in 20 ml DI-water, poured into the suspension. After stirring for 10 min, and ultra-sounding for 60 min, pouring into a high-pressure kettle, at 120 °C for 12 h. Finally, freeze-dried for 72 h.

Material characterization. X-ray diffraction (XRD) (PANalytical) patterns were tested on an 8 Kev Cu Ka radiation diffractometer. Scanning electron microscopy (SEM) images were conducted on Tescan/Clara microscope. Raman spectroscopy was carried out on a Renishaw/inVia spectrometer.

Electrochemical measurement. Zn/Zn symmetric cells, Zn/Ti half cells, and $Zn/MnO_2$ full cells were assembled for cycling tests in an incubator with a temperature of 25 °C. Cathodes were prepared by mixing PVDF, super P, and $MnO_2$ in a mass ratio of 1:2:7, and stirring for 12 h after adding NMP. The produced slurry was coated on stainless steel with active material ($MnO_2$) loading of 0.8–1 mg $cm^{-2}$ or 4.0 mg $cm^{-2}$. Then the stainless steel coated with the active material was dried at 80 °C overnight in a vacuum oven and was punched into small round pieces of 1 cm in diameter. The specific capacity was calculated based on the mass of the active material $MnO_2$. The zinc metal foils with a thickness of 0.1 mm were punched into small round pieces of 1.2 cm in diameter to serve as anodes. Glass fiber separators (GF/D) with a diameter of 1.9 cm were severed as the separators. All of the coin cells were assembled with 400 μl electrolyte and studied on battery testing instruments (Land, China). All the full cells were tested in the range of 0.8–2.0 V at various current densities from 0.5 to 5 A $g^{-1}$. Both electrochemical impedance spectroscopy (EIS) and cyclic voltammetry (CV) were tested on Gamry electrochemical stations. CV tests were recorded at a scanning rate of 1 mV $s^{-1}$ from 0.8 V to 2.0 V. EIS was collected at a frequency of 0.01 Hz–100 kHz.

Theoretical calculations. All atomistic simulations were performed using GROMACS package with cubic periodic boundary conditions[50]. The equations for the motion of all atoms were integrated using a classic Verlet leapfrog integration algorithm with a time step of 1.0 fs. A cutoff radius of 1.6 nm was set for short-range van der Waals interactions and real-space electrostatic interactions. First-principles calculations were performed using the Vienna ab initio Simulation Package (VASP 3.5.3) with density functional theory (DFT)[51]. The Perdew-Bruke-Ernzerh of exchange-correlation functional of the generalized-gradient approximation (GGA) was adopted, and the cutoff energy for this plane-wave basis set was set to be 450 eV, and the Γ-centered k-point grids were used for Brillouin zone integrations. The exchange-correlation functional with a Gaussian smearing width term of 0.05 eV was used. The convergence criterion for electronic self-consistent iteration was set to $1 \times 10^{-5}$ eV. The $SrTiO_3$ (100), (110), and (111) electrode surface was constructed from optimized primitive cell and are consisting of 4 × 4, 4 × 3, and 3 × 3 primitive cells consisting of at least four atom layers. The atoms in the top two layers were free to simulate surface state, and the atoms in the other layers were fixed during calculation to simulate bulk $SrTiO_3$ structures. A vacuum of 15 Å was contained in each modeling system to reduce interactions between each surface. All structures were fully relaxed to their optimized geometries with the force convergence set to 0.01 eV/Å.

## Reporting summary

Further information on research design is available in the Nature Portfolio Reporting Summary linked to this article.

## Data availability

The authors declare that the data supporting the findings of this study are available within the paper and its Supplementary Information files. Should any raw data files be needed in another format they are available from the corresponding author upon reasonable request. Source data are provided with this paper.

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

## Acknowledgements

The authors gratefully acknowledge the National Key R&D Program of China (Grant No. 2022YFB2404600), National Natural Science Foundation of China (Grant No. 52172264) and the Natural Science Foundation of Hunan Province of China (Grant No. 2021JJ10060 and No. 2022GK2033).

## Author contributions

F.W. conceived ideas, designed experiments and edited article drafts. R.D. executed and analyzed all the main electrochemical experiments and wrote original draft. Z.H. reviewed and edited article drafts. F.C. helped with the electrochemical tests. J.L. performed zinc electrodeposition and SEM studies. Y.C. and Y.Z. edited article drafts.

## Competing interests

The authors declare no competing interests.
