## [Peer Review File · Nature Communications]

An aqueous electrolyte densified by perovskite SrTiO₃ enabling high-voltage zinc-ion batteriesREVIEWER COMMENTS

Reviewer #1 (Remarks to the Author):

In this article, the authors present a new densified electrolyte adding a perovskite material (SrTiO₃) to a ZnSO₄ solution. This way of preparing the electrolyte seems like an original idea and the results seem to corroborate the good properties of this electrolyte. When analyzing the deposition of Zn²⁺ cations on this material, it seems that the formation of dendrites is avoided, which is an important milestone to obtain aqueous Zn-based rechargeable batteries. Furthermore, several theoretical and experimental techniques have been used to characterize the electrolyte.

However, there are diverse aspects that need to be corrected, as it is commented below:

- 1) Authors should explain why 50 wt.% SrTiO₃ is used in the electrolyte. Have experiments been done with other percentages of perovskite? How does the amount of perovskite affect the electrolyte?
- 2) In the article I miss measurements of ionic conductivity of the electrolyte. These should be included. Also, viscosity measurements would be nice.
- 3) The calculation of the transfer number is wrong. The authors cite reference [38] and the text follows what that reference says, but the calculation seems to be done with the EIS resistances measured "before" and "after" the polarization (according to the data in figures 3g and h). With these values the $t_{zn^{2+}}$ can be calculated following the Bruce-Vicent-Evans' method ([doi.org/10.1016/0032-3861\(87\)90394-6](https://doi.org/10.1016/0032-3861(87)90394-6) or doi.org/10.1016/j.jallcom.2023.168877). It would be nice to compare the results of both methods. Please, in the method of reference 38 indicate the voltage used, as well as the initial and final intensities.
- 4) With respect to the specific current and capacity values, authors should indicate what is the electrode dimensions (cm²) and what mass is used to mA_hg⁻¹ values. Only 0.8-1 mg cm⁻² of MnO₂? is indicated in the method section.
- 5) Considering the mass used is that of MnO₂ (1 mg) the capacity of the batteries (around 320 mA_hg⁻¹ at 1 A g⁻¹ in Figure 6b) would be 0.3 mA_h at 1 mA, which are not high values.
- 6) Authors must confirm the reproducibility of the experiments. The specific capacity of Figure 6a (300 mA_hg⁻¹ at 1A g⁻¹, after 100 cycles) does not coincide with the result observed in Figure 6c, where 400 mA_hg⁻¹ is reached.
- 7) Regarding Zn/MnO₂ voltammograms, it is indicated that double redox pairs are obtained and that the presence of SrTiO₃ does not influence in the redox behavior. However, the oxidation peaks, as well as the cathodic peak at 1.4 V, grow with SrTiO₃, while the cathodic peak at 1.2 V does not change. Therefore, there is indeed a change in behavior and the authors should try to explain it.
- 8) Pag. 10 (paragraph from 274-293). It is included "...the overpotential of the cell using the densified electrolyte is slightly larger than that of the cell using conventional electrolyte, which can be ascribed to the larger potential barrier that the Zn (002) plane formation process needs to overcome. This can be further demonstrated by the fact that when raising the current density to 2 mA cm⁻² with 1 mA_h cm⁻²..."
this claim does not hold up, especially considering that figure S10 contradicts it. This seems to be related to the non-repetitiveness of the results.
- 9) The figures on DFT calculations must specify well the colors and the species absorbed, like H₂O in S3.
- 10) There are some typos throughout the entire text. Such as, uA_hcm⁻² (pag 8, twice); Fig 4i and 4j (pag 9); etc.

Reviewer #2 (Remarks to the Author):

In this manuscript, the authors present an innovative aqueous electrolyte design for zinc ion batteries. By incorporating SrTiO₃ particles into the conventional electrolyte, the authors were able to expand the thermodynamic stability window of the aqueous electrolytes and achieve non-dendritic Zn deposition with a preferred orientation of (002). Extensive experiments have been carried out to support the design. The results could be of general interest to the electrochemistry and battery community. While finding the results interesting, I still think certain analysis/interpretation remain to be revisited and revised with extra care. I would recommend publication of the manuscript if the authors could fully address my concerns. Below, I have detailed my questions and some minor comments,

1. The choice of SrTiO₃ as the additive seems to be an interesting one. Why SrTiO₃ was chosen in this study? What would be the design/screening principle for such inorganic additives? This is important to enlighten future study in a systemic way.
2. For the name “densified electrolyte”, I am not sure “densified” is an appropriate term to describe the new electrolyte, rather it is misleading to some degree. I kept thinking which part of the electrolyte gets densified. (But in respect to the authors’ will and for the sake of simplicity, I will keep use “densified” in the following comments.)
3. In the Zn-Zn symmetric cell test, the overpotential in densified electrolyte becomes larger over cycling. The increasing overpotential is common in Li metal symmetric cell tests due to the build-up of residue SEI and depletion of electrolyte. But there should be no SEI on Zn anode as the authors have indicated. What could be the cause of this overpotential growth in Zn-Zn symmetric cells for densified electrolyte? From my perspective, an opposite trend in the overpotential (keep decreasing or at least remain constant) seems more reasonable, as Zn anode would have increased surface area upon repetitive deposition and dissolution.
4. Closely coupled with my question 3, the stripping curve of ZnSO₄-SrTiO₃-500th in Figure 5h indicates a similar polarized behavior. What could be the origin? Is Zn anode surface still “SEI” free in this new electrolyte?
5. Figure 5d and 5e, two different current densities and deposition capacities were investigated. I failed to find consistency in electrolyte ionic conductivity here. The overpotential of densified electrolyte at 2 mA/cm² is lower than conventional electrolyte, while the overpotential at 1 mA/cm² is higher than conventional electrolyte. Given that no SEI on Zn anode, and from Figure 3, interfacial charge transfer resistance in densified electrolyte is lower than that of conventional electrolyte, these results suggest that the electrolyte ionic conductivity has non-trivial variation. Could the authors measure the ionic conductivity of both electrolytes and compare?
6. Was the same amount of liquid portion of the electrolyte used or the same total weight of electrolyte used in all these measurements?
7. In Zn/MnO₂ full cell with conventional electrolyte (2M ZnSO₄, 0.1M MnSO₄), the redox reaction is a single-electron redox reaction (Mn(IV) - Mn(II)). The theoretical specific capacity of MnO₂ in single-electron redox reaction is 308 mAh/g. However, in densified electrolyte, the reported specific capacities apparently exceed this value, seems to suggest the

coexistence of both single-electron redox and two-electron redox (Mn(IV) - Mn(II)) reactions. The two-electron pathway corresponding to MnO₂-Mn²⁺(aq) conversion only happens at quite acidic condition and does not seem possible in the densified electrolyte. I suspect that part of the Mn²⁺ from the electrolyte (0.1M MnSO₄) got oxidized into MnO₂ since in densified electrolyte the cell was charged up to 2 V, therefore effectively increase the actual MnO₂ loading. I would recommend the authors to check cycling behavior in electrolytes without 0.1M MnSO₄. If such high specific capacity still exists, the authors might need to explain the mechanism of the potential two-electron redox in the densified electrolyte.

Minor comments:

1. In Figure 5g, the two electrolytes are mislabeled.
2. When the authors reported specific capacity, I assume that it is calculated based on MnO₂ cathode, but please clarify the specific capacity of what in the manuscript.
3. Page 4, line 114, the authors indicate that the densified electrolyte has “good mechanical strength”. Is this property relevant to Zn metal anode or full cell performance? I did not find either measurement on this property or clear logical relation in the later discussion with the battery performance.

Point-by-point response

Reviewer #1:

In this article, the authors present a new densified electrolyte adding a perovskite material (SrTiO_3) to a ZnSO_4 solution. This way of preparing the electrolyte seems like an original idea and the results seem to corroborate the good properties of this electrolyte. When analyzing the deposition of Zn^{2+} cations on this material, it seems that the formation of dendrites is avoided, which is an important milestone to obtain aqueous Zn-based rechargeable batteries. Furthermore, several theoretical and experimental techniques have been used to characterize the electrolyte.

However, there are diverse aspects that need to be corrected, as it is commented below:

Answer: Dear Reviewer #1, thank you very much for evaluating our manuscript and appreciating our work. We have explained all the questions and adjusted the manuscript per your suggestions.

1) Authors should explain why 50 wt.% SrTiO_3 is used in the electrolyte. Have experiments been done with other percentages of perovskite? How does the amount of perovskite affect the electrolyte?

Answer: Thank you for your very useful suggestion. We have done the experiments with various percentages of SrTiO_3 . As shown in **Supplementary Fig. S4**, the degree of H-bond destruction of the electrolyte is more serious with the increase of SrTiO_3 particles. Besides, the mass ratio of SrTiO_3 has a crucial effect on the cycle stability of Zn/ MnO_2 cells, and 50 wt.% as the most suitable ratio demonstrates the best cycle stability and capacity retention (**Supplementary Fig. S14**).

Revised manuscript:

“To further investigate the variation of the H-bond network structure with SrTiO_3 content, Raman spectra of electrolytes with different SrTiO_3 contents (0 ~ 60 wt.%) and their corresponding fitting peaks were analyzed. As shown in Fig. 1h, the strong H-bond located at 3253 cm^{-1} exhibits a clear downward trend as the content increases. When the content is up to 60 wt.%, the proportion of strong H-bond is only 34 %, which

indicates that more SrTiO₃ particles can damage more H-bond network of the electrolyte (Supplementary Fig. S4).”

“To investigate the effect of SrTiO₃ content on the electrochemical performance, Zn/MnO₂ full cells using conventional electrolyte with various SrTiO₃ contents were assembled. As shown in Supplementary Fig. S14, the cells using conventional electrolyte shows a drastic decay, with a capacity retention rate of 43 % after 500 cycles at a current density of 1 A g⁻¹. In contrast, densified electrolytes can significantly improve the specific capacity and more SrTiO₃ particles are beneficial to the capacity retention. However, excessive content (60 wt.%) leads to an excessive densification with visible reductions of mobility and conductivity, leading to poor electrochemical performance. The cell with 50 wt.% SrTiO₃ exhibits the best cycle stability in the densified electrolytes, demonstrating an initial specific capacity of 278.8 mA h⁻¹ and a slightly increased specific capacity of 328.2 mAh g⁻¹ at 500th cycle (Fig. 6a).”

Fig. 1 | The properties of aqueous densified electrolyte. a The crystal structure of SrTiO₃. **b** The XRD spectra of the SrTiO₃ powder and its standard PDF card. **c** Schematic diagram of densified aqueous electrolytes. **d** Schematic illustration of densified electrolytes formed by increasing density of solution after the addition of oxide. **e** Comparison of Raman spectra of different electrolytes. Raman fit peaks of **f** ZnSO₄ electrolyte and **g** aqueous densified electrolyte. **h** The ratio of fitting strong H-bond area of electrolytes with various SrTiO₃ contents. **i** The Top view of geometrical configurations of H₂O adsorbed on the Ti atom of SrTiO₃ (110) plane. The boxes of molecular dynamics simulations with main solvated structure of Zn²⁺ in **j** ZnSO₄

electrolyte and **k** aqueous densified electrolyte.

Supplementary Fig. S4 | Raman spectra with fit peaks of different electrolytes: **a** ZnSO₄ electrolyte, **b** ZnSO₄ electrolyte with 20 wt.% SrTiO₃, **c** ZnSO₄ electrolyte with 40 wt.% SrTiO₃, **d** ZnSO₄ electrolyte with 50 wt.% SrTiO₃, **e** ZnSO₄ electrolyte with 60 wt.% SrTiO₃; **f** The ratio of strong H-bond area of electrolytes with various SrTiO₃ contents.

Supplementary Fig. S14 | The long-term galvanostatic cycling performance of Zn/MnO₂ cells in electrolytes with different SrTiO₃ contents at a current density of 1 A g⁻¹.

2) In the article I miss measurements of ionic conductivity of the electrolyte. These should be included. Also, viscosity measurements would be nice.

Answer: In this revision, we have tested the conductivity of the electrolyte with an ionic conductivity instrument and the viscosity with a rotational rheometer as you suggested.

Revised manuscript: “The conductivity of the densified electrolytes is slightly smaller than that of the conventional electrolyte, and gradually decreases with the increase of SrTiO₃ content, which is mainly due to the fact that SrTiO₃ is an insulating material and the viscosity of the densified electrolyte is higher than that of the conventional electrolyte (Supplementary Fig. S3).”

Supplementary Fig. S3 | a The conductivities of electrolytes with different SrTiO₃ contents. **b** their corresponding viscosities tested on a rotational rheometer at a shear rate of 1000 s⁻¹.

3) The calculation of the transfer number is wrong. The authors cite reference [38] and the text follows what that reference says, but the calculation seems to be done with the EIS resistances measured "before" and "after" the polarization (according to the data in Fig.s 3g and h). With these values the $t_{zn^{2+}}$ can be calculated following the Bruce-Vicent-Evans' method ([doi.org/10.1016/0032-3861\(87\)90394-6](https://doi.org/10.1016/0032-3861(87)90394-6) or doi.org/10.1016/j.jallcom.2023.168877). It would be nice to compare the results of both methods. Please, in the method of reference 38 indicate the voltage used, as well as the initial and final intensities.

Answer: Thank you for your very helpful suggestion. After careful investigation of the

references related to the calculation of transference number, we believe that the Bruce-Vicent-Evans' method is more universal. In addition, we reduced the polarization voltage from the original 20 mV to 10 mV according to the first reference to reduce the error. We have also compared the results of the two calculation methods as shown below according to your suggestions (**Supplementary Fig. S27**). It can be found that different results can be obtained by using two different calculation methods. Considering that the Bruce-Vicent-Evans' method adds a correction term, we believe that the corresponding calculated results are more reliable. And we have modified Fig. 3g and 3h in the manuscript.

Revised manuscript: “The equations calculated the Zn^{2+} transference number as follows:^{38, 39}

$$t_{\text{Zn}^{2+}} = I_s(\Delta V - I_0 R_0) / I_0(\Delta V - I_s R_s)$$

Where R_0 is the resistance before polarization, R_s is resistance after polarization, ΔV is the polarization potential, I_0 is the initial current, and I_s is the stable state current. The symmetric cell using 2 M ZnSO_4 electrolyte delivers a low value of 0.42 (Fig. 3g). The transference numbers are significantly increased by adding SrTiO_3 , and the obtained number increase to 0.64 with a significant decreased charge transfer resistance (Fig. 3h). These results indicate that the densified electrolyte has a prominent performance in broadening the electrochemical window, promoting the interface stability, and improving Zn^{2+} transference number.”

Fig. 3 | Chemical and electrochemical properties of two electrolytes. a Electrochemical stability window. **b** Corresponding zoomed-in curves of tiny areas. **c** The linear sweep voltammetry (LSV) for test of HER. **d** Tafel curves in three-electrode cells. XRD patterns of Zn foils soaked in **e** ZnSO₄ electrolyte and **f** the densified electrolyte. The Zn²⁺ transference number measured by electrochemical polarization at a small voltage of 10 mV in **g** ZnSO₄ electrolyte and **h** the densified electrolyte.

“38. James E, Colin A. V, Peter G. B. Electrochemical measurement of transference numbers in polymer electrolytes. *Polymer*. **28**, 2324-2328 (1987).

39. Wang T, *et al.* 3D nanofiber framework based on polyacrylonitrile and siloxane-modified Li_{6.4}La₃Zr_{1.4}Ta_{0.6}O₁₂ reinforced poly (ethylene oxide)-based composite solid electrolyte for lithium batteries. *J. Alloys Compd.* **945**, 168877 (2023).”

Supplementary Fig. S27 | Comparison results of the Zn^{2+} transference numbers obtained by two calculation methods: a) the Bruce-Vicent-Evans' method and b) the original method ($t = R_{cell}/R_{DC}$) for conventional electrolyte; c) the Bruce-Vicent-Evans' method and d) the original method for the densified electrolyte. The Zn^{2+} transference numbers were measured by electrochemical polarization at a small voltage of 10 mV.

4) With respect to the specific current and capacity values, authors should indicate what is the electrode dimensions (cm^2) and what mass is used to $mAhg^{-1}$ values. Only $0.8-1 mg cm^{-2}$ of MnO_2 ? is indicated in the method section.

Answer: Thank you for your kind suggestion. The diameter of the electrode was 1 cm. The mass of MnO_2 was used to calculate the specific capacities. We also provided more studies on thick electrodes ($4.0 mg cm^{-2}$) in this revision. And according to your suggestion, we have added more information into the method section.

Revised manuscript: “Cathodes were prepared by mixing PVDF, super P, and MnO_2 in a mass ratio of 1:2:7, and stirring for 12 hours after adding NMP. The produced slurry was coated on stainless steel with active material (MnO_2) loading of $0.8-1 mg cm^{-2}$ or $4.0 mg cm^{-2}$. Then the stainless steel coated with the active material was dried at $80 \text{ }^\circ C$

overnight in a vacuum oven and was punched into small round pieces of 1 cm in diameter. The specific capacity was calculated based on the mass of the active material MnO_2 .”

5) Considering the mass used is that of MnO_2 (1 mg) the capacity of the batteries (around 320 mAh g^{-1} at 1 A g^{-1} in Fig. 6b) would be 0.3 mAh at 1 mA, which are not high values.

Answer: As you suggested, we have assembled Zn/ MnO_2 full cells using cathodes with a higher mass loading. In **Supplementary Fig. S21**, the cathode with a high active mass loading of 4 mg cm^{-2} delivers a specific capacity of 238 mAh g^{-1} and a capacity retention rate of 95% after 200 cycles at 0.5 A g^{-1} , which increases the capacity from 0.3 mAh to 0.9 mAh.

Revised manuscript: “To be more practical, full cells with a high mass loading of 4 mg cm^{-2} were studied in Supplementary Fig. S21. The cell using densified electrolyte delivers a specific capacity of 238 mAh g^{-1} after 200 cycles at 0.5 A g^{-1} with stable CEs. In contrast, the cell using conventional electrolyte only exhibits a low specific capacity and fails in 170th cycle.”

Supplementary Fig. S21 | The galvanostatic cycling performance of Zn/ MnO_2 cells with a higher-loading of 4 mg cm^{-2} in various electrolytes at a current density of 0.5 A g^{-1} .

6) Authors must confirm the reproducibility of the experiments. The specific capacity of Fig. 6a (300 mAh g^{-1} at 1 A g^{-1} , after 100 cycles) does not coincide with the result observed in Fig. 6c, where 400 mAh g^{-1} is reached.

Answer: Thank you for your kind suggestion. We confirm the reproducibility of the experiments. In fact, the cells shown in Fig. 6a are directly charged and discharged at 1 A g^{-1} without any activation process by using a smaller current density, therefore, it does not coincide with the result in Fig. 6c. In order to further provide the effect of activation process, we did a new test to confirm the reproducibility in this revision. The Zn/MnO₂ cell after ten cycles of activation at 0.5 A g^{-1} demonstrated a specific capacity around 400 mAh g^{-1} (**Supplementary Fig. S22**).

Revised manuscript: More importantly, when the applied current density returns to 1 A g^{-1} , the specific capacity of the cell using densified electrolyte recovers to 392.6 mAh g^{-1} . The higher specific capacity in Fig.6c than the capacity in Fig.6a is due to the presence of an electrochemical activation process at a relatively smaller current density, which is further proved in the Supplementary Fig. S22. In contrast, the specific capacity only recovers to 129.9 mAh g^{-1} in conventional electrolyte.

Revised supporting information: “The higher specific capacity may be due to the presence of an activation process, which leads to more zinc-manganese oxide generation and thus contributes more specific capacity (Supplementary Fig. S22). And this is the reason why the specific capacity of the cell using densified electrolyte recovers to 392.6 mAh g^{-1} , which is higher than that of the cell before rate tests and the cells in Fig. 6a that cycle directly at 1 A g^{-1} , when the applied current density returns to 1 A g^{-1} .”

Supplementary Fig. S22 | The electrochemical performance of the cell using densified electrolyte after ten cycles of activation at 0.5 A g⁻¹.

7) Regarding Zn/MnO₂ voltammograms, it is indicated that double redox pairs are obtained and that the presence of SrTiO₃ does not influence in the redox behavior. However, the oxidation peaks, as well as the cathodic peak at 1.4 V, grow with SrTiO₃, while the cathodic peak at 1.2 V does not change. Therefore, there is indeed a change in behavior and the authors should try to explain it.

Answer: Combining your suggestion with the comments of reviewer#2, we did find that SrTiO₃ changed the electrochemical behavior of Zn/MnO₂ full cells. To exclude the interference of the cathode material, we designed an experiment to carry out cyclic voltammetry test using the cathode composed of Super P and PVDF without any MnO₂ (defined as SP cells). As shown in **Supplementary Fig. S19**, the CV curves of the cells using conventional electrolytes and the densified electrolytes exhibit significantly different electrochemical behaviors. We have found that there is an additional redox reaction ($x\text{Zn}^{2+} + y\text{Mn}^{2+} + \text{H}_2\text{O} \rightleftharpoons \text{Zn}_x\text{Mn}_y\text{O} + 2\text{H}^+ + (2-2x-2y)e^-$) at the cathodic peak around 1.4 V as well as at the corresponding anodic peak, which can be enhanced by the addition of SrTiO₃.

Firstly, we studied the electrochemical performance of Zn/MnO₂ full cells without 0.1M MnSO₄. It is found that the electrochemical performances of the cells are significantly

decreased, which indicates that MnSO_4 plays an indispensable role in both electrolytes (**Supplementary Fig. S16a**). To verify whether the Mn^{2+} in the densified electrolyte is oxidized into MnO_2 , we studied the performance of Zn/MnO_2 full cells without ZnSO_4 , in which only the insertion of H^+ into MnO_2 is involved. As shown in **Supplementary Fig. S16b**, the performance of the cell using densified electrolyte is even worse than conventional electrolyte. This suggests two things: First, the hypothesis that the Mn^{2+} in the densified electrolyte is oxidized into MnO_2 may not be correct; Second, the H^+ content of the densified electrolyte is lower than that of the conventional electrolyte, that is, the pH of the densified electrolyte may be higher than that of conventional electrolyte.

Next, we tested the pH of conventional electrolyte and densified electrolytes with various mass ratios of SrTiO_3 . **Supplementary Fig. S15** demonstrates that once SrTiO_3 is added, the pH values of the densified electrolytes increase rapidly to more than 5.5, and the pH of the densified electrolyte with 50 wt.% increases from 3.4 to 5.76. This may be due to the fact that SrTiO_3 is an alkaline oxide, which inhibits hydrolysis of the densified electrolytes and thus increases the pH values.

Then, we hypothesized that the exceeded capacity may be due to additional electrochemical reaction that occurred after the pH increase. According to some published article (10.1002/adma.202300053, 10.1039/D1EE03547A, 10.1016/j.ensm.2019.12.021), when pH is around 6, in addition to the co-insertion of $\text{H}^+/\text{Zn}^{2+}$, the reaction ($x\text{Zn}^{2+} + y\text{Mn}^{2+} + \text{H}_2\text{O} \rightleftharpoons \text{Zn}_x\text{Mn}_y\text{O} + 2\text{H}^+ + (2-2x-2y)e^-$) also occurs. We designed an experiment to test and verify this hypothesis. The cathode composed of Super P and PVDF without any MnO_2 was used, and the produced cell was defined as SP cell. **Supplementary Fig. S17a** shows the performance of the SP cell using conventional electrolyte (2M ZnSO_4 , 0.1M MnSO_4). When the charging cutoff voltage is 1.8 V, the cell has almost no capacity. When the voltage is raised to 2 V, the specific capacity is about 15 mAh g^{-1} (we set the active material mass to 1 mg). The SP cell using densified electrolyte exhibits a low specific capacity of 20 mAh g^{-1} under the charge cut-off voltage of 1.8 V, but deliver a higher specific capacity about 100 mAh g^{-1} under voltage of 2.0 V. (**Supplementary Fig. S17b**). This indicates that

increasing the voltage to 2.0 V is an essential condition for obtaining high capacity. More importantly, we found that there is significantly difference in the charge-discharge curves. The charge-discharge curves of the first two circles of the SP cell using conventional electrolyte have distinct plateaus around 1.99V (**Supplementary Fig. S17c**), which is caused by the oxidation of small amounts of Mn^{2+} to MnO_2 (10.1002/anie.201904174). In contrast, the cell with densified electrolyte provides a longer platform below 1.7 V, and the curve is slowly raised to 2.0 V, indicating that there is almost no electrochemical behavior of Mn^{2+} oxidation to MnO_2 at high voltage plateaus. This is consistent with the charge-discharge curves in the whole Zn/ MnO_2 full cells (**Supplementary Fig. S18**), which indicates that the electrochemical behavior in conventional electrolyte is indeed different from that in densified electrolyte.

To further analyze the electrochemical behaviors, the cyclic voltammetry tests of SP cells were carried out at a scan rate of 0.1 mV s^{-1} (**Supplementary Fig. S19**). Since the cathode contains only SP without any MnO_2 , the first discharge curve does not have any current. During the first charge, the SP cell using conventional electrolyte has a sharp peak at around 2.0 V, which leads to the generation of MnO_2 . In the second discharge curve, the SP cell with conventional electrolyte shows a similar peak to that of the Zn/ MnO_2 full cells, that is, a smaller peak around 1.4 V and a larger peak around 1.2 V. In the second charge curve, the SP cell using conventional electrolyte still shows a similar peak to that of the Zn/ MnO_2 full cells, and with a sharp peak at 2.0 V to produce MnO_2 . This is further evidenced by the third cycle of the discharge curve in the conventional electrolyte, where the peak currents at both 1.4 V and 1.2 V increase, indicating more MnO_2 will be still produced on once charging to 1.99 V (**Supplementary Fig. S19a**). As a contrast, there is only a small peak at 2.0 V, indicating trace amounts of MnO_2 could be produced at 2.0 V in the first charge, while a special peak at 1.6 V in densified electrolyte is also visible, corresponding to $\text{Zn}_x\text{Mn}_y\text{O}$ production (**Supplementary Fig. S19b**). The 1.4 V peak is significantly larger than the 1.2 V peak in the next discharge curve (**Supplementary Fig. S19b**), which is consistent with the CV curves of Zn/ MnO_2 full cells (**Fig. 6e**), indicating that $\text{Zn}_x\text{Mn}_y\text{O}$ does generate in the previous process. More importantly, the SP cells using

densified electrolyte only show one peak at 1.65V without any peak close to 2.0 V, suggesting that only a small amount of MnO₂ is produced in the first charge, and there would be no any MnO₂ produced after the second cycle. Furthermore, the densified electrolyte remains almost unchanged during the third charge curve, but the conventional electrolyte shows an additional peak at 1.65 V, which may be due to the increased pH of the conventional electrolyte leading to Zn_xMn_yO production (10.1038/s41467-022-29987-x, 10.1002/sml.202005406).

Finally, we conducted XRD tests on the cathode of the SP cells charged to 2.0 V for the first time. In the conventional electrolyte, we only detected the presence of the phase of the stainless steel, and did not detect MnO₂, which may be because the amount is too small to be detected (**Supplementary Fig. S20a and S20b**). In contrast, in addition to the peak for stainless steel and SrTiO₃ in the densified electrolyte, we also found some additional peaks between 26 and 30 degrees, which are attributed to Zn_xMn_yO (**Supplementary Fig. S20c and S20d**).

In summary, all of these results demonstrate the existence of an electrochemical reaction in densified electrolytes, that is: $x\text{Zn}^{2+} + y\text{Mn}^{2+} + \text{H}_2\text{O} \rightleftharpoons \text{Zn}_x\text{Mn}_y\text{O} + 2\text{H}^+ + (2-2x-2y)\text{e}^-$. And we have revised the manuscript according to your suggestions.

Revised manuscript: “It is worth mentioning that the specific capacity of Zn/MnO₂ full cells with densified electrolyte has exceeded the theoretical value of 308 mAh g⁻¹. This is because the electrolyte pH increased from 3.4 to 5.76 after the addition of SrTiO₃ (Supplementary Fig. S15), resulting in additional electrochemical reaction ($x\text{Zn}^{2+} + y\text{Mn}^{2+} + \text{H}_2\text{O} \rightleftharpoons \text{Zn}_x\text{Mn}_y\text{O} + 2\text{H}^+ + (2-2x-2y)\text{e}^-$).^{43, 44, 45} To exclude the interference of the active material MnO₂, the cathode containing only Super P and PVDF is designed to assemble the cell (note as SP cell). It is found that SP cells can only make additional capacity contributions when the densified electrolytes contain MnSO₄, ZnSO₄, SrTiO₃ and is charged to 2.0 V (Supplementary Fig. S16 and S17). The charge-discharge curves of the SP cell using conventional electrolyte have distinct plateaus around 1.99 V, which is caused by the oxidation of small amounts of Mn²⁺ to MnO₂.⁴⁶ In contrast, the cells with densified electrolytes have a longer platform below 1.7 V, and the curve is slowly

raised to 2.0 V. This is consistent with the charge-discharge curves in the whole Zn/MnO₂ full cells (Supplementary Fig. S18), which indicates that the electrochemical behavior of conventional electrolyte is indeed different from that of densified electrolyte. The cyclic voltammetry tests of the SP cells were carried out to further study the electrochemical behavior. As shown in Supplementary Fig. S19a, after the first charge to 2.0 V, the CV curves of SP cell with conventional electrolyte is almost the same as that of Zn/MnO₂ full cells, indicating that Mn²⁺ is oxidized to MnO₂. Peculiarly, a weak peak corresponding to Zn_xMn_yO can be observed at the third cycle of the CV curve, which is because the pH increases due to side reactions, further indicating that the Zn_xMn_yO would be produced in electrolytes with higher pH^{47, 48}. In contrast, the densified electrolyte showed no sharp peaks at 2.0 V indicating almost no MnO₂ production, but distinct peaks at 1.65 V and 1.35 V, which correspond to the redox reaction of Zn_xMn_yO (Supplementary Fig. S19b). The XRD patterns of cathodes of SP cells after charging to 2.0 V provides further evidence (Supplementary Fig. S20). In the conventional electrolyte, the XRD pattern only shows that the presence of the stainless steel but without MnO₂, which may be due to the amount is too small to be detected. In contrast, in addition to the peak for stainless steel and SrTiO₃ in the densified electrolyte, the XRD pattern exhibits several peaks, which are attributed to Zn_xMn_yO. Overall, all the results show that the pH can be stabilized at about 5.8 after the addition of SrTiO₃, which leads to a continuous and reversible reaction: $x\text{Zn}^{2+} + y\text{Mn}^{2+} + \text{H}_2\text{O} \rightleftharpoons \text{Zn}_x\text{Mn}_y\text{O} + 2\text{H}^+ + (2-2x-2y)\text{e}^-$.

“The cyclic voltammetry (CV) profiles were collected at a scan rate of 1 mV s⁻¹ to verify the electrochemical behaviors. Zn/MnO₂ full cells using both conventional electrolyte and the densified electrolyte exhibit similar shapes with two couples of redox peaks, but the cathodic peak at 1.35 V significantly increases in densified electrolytes compared to conventional electrolytes, while the cathodic peak at 1.2 V does not change, which is due to the additional redox reaction contributing to the capacity ($x\text{Zn}^{2+} + y\text{Mn}^{2+} + \text{H}_2\text{O} \rightleftharpoons \text{Zn}_x\text{Mn}_y\text{O} + 2\text{H}^+ + (2-2x-2y)\text{e}^-$). Besides, the battery using the densified electrolyte exhibits relatively higher response current and reduction potential, indicating faster reaction kinetics, lower overpotential, and better reversibility. (Fig.

6e).”

“43. Yang H, *et al.* Protocol in Evaluating Capacity of Zn–Mn Aqueous Batteries: A Clue of pH. *Adv. Mater* DOI: 10.1002/adma.202300053 (2023).

44. Yang H, *et al.* The origin of capacity fluctuation and rescue of dead Mn-based Zn–ion batteries: a Mn-based competitive capacity evolution protocol. *Energy Environ. Sci.* **15**, 1106-1118 (2022).

45. Vaiyapuri S, *et al.* The dominant role of Mn^{2+} additive on the electrochemical reaction in $ZnMn_2O_4$ cathode for aqueous zinc-ion batteries. *Energy Stor. Mater.* **28**, 407-417 (2019).

46. Chao D, *et al.* An Electrolytic Zn-MnO₂ Battery for High-Voltage and Scalable Energy Storage. *Angew. Chem., Int. Ed.* **58**, 7823-7828 (2019).

47. Yangmoon K, *et al.* Corrosion as the origin of limited lifetime of vanadium oxide-based aqueous zinc ion batteries, *Nat. Commun.* **13**, 2371 (2022).

48. Sung J K, *et al.* Unraveling the Dissolution-Mediated Reaction Mechanism of α -MnO₂ Cathodes for Aqueous Zn-Ion Batteries, *Small.* **16**, 2005406 (2020).”

Supplementary Fig. S15 | The pH of conventional electrolyte and densified electrolytes with different SrTiO₃ contents.

Supplementary Fig. S16 | The galvanostatic cycling performance of Zn/MnO₂ cells at a current density of 1 A g⁻¹ in various electrolytes: **a** without MnSO₄ and **b** without ZnSO₄.

Supplementary Fig. S17 | The galvanostatic cycling performance of Zn/SP cells (set the active material mass is 1 mg) under different charge cut-off voltage at a current density of 1 A g⁻¹ in **a** conventional electrolytes and **b** densified electrolyte; The first two charge-discharge curves of SP cells under charge cut-off voltage of 2.0 V in **c** conventional electrolytes and **d** densified electrolytes.

Supplementary Fig. S18 | The first two charge and discharge curves of Zn/MnO₂ full cells in **a** conventional electrolyte and **b** densified electrolyte.

Supplementary Fig. S19 | The cyclic voltammetry (CV) profiles of SP cells at a scan rate of 0.1 mV s⁻¹: **a** in the conventional electrolyte and **b** in the densified electrolyte.

Supplementary Fig. S20 | The investigation on the redox behavior of the first charge to 2.0 V: **a** the XRD pattern and **b** the corresponding local magnification ranges (marked area) of the SP cathode charged in the conventional electrolyte; **c** the XRD pattern and **d** the corresponding local magnification ranges (marked area) of the SP cathode charged in the densified electrolyte.

8) Pag. 10 (paragraph from 274-293). It is included “...the overpotential of the cell using the densified electrolyte is slightly larger than that of the cell using conventional electrolyte, which can be ascribed to the larger potential barrier that the Zn (002) plane formation process needs to overcome. This can be further demonstrated by the fact that when raising the current density to 2 mA cm⁻² with 1 mAh cm⁻²...”. this claim does not hold up, especially considering that Supplementary Fig. S10 contradicts it. This seems to be related to the non-repetitiveness of the results.

Answer: Thank you for point out the mistake. After careful check on the data, sure enough, we got it wrong. We are very sorry that we accidentally reversed the data when

we were processing and drawing graphic. We negligently reversed the data of densified electrolytes in Fig. 5d and 5e, that was, we put the data of 1 mA cm^{-2} of densified electrolyte in Fig. 5e (represent 2 mA cm^{-2}), but put the data of 2 mA cm^{-2} in Fig. 5d, which led to the inconsistency of the results. Now, the correct data have been updated in Fig. 5. Meanwhile, to ensure the reliability of the data, we repeated several new experiments in this revision to compare the overpotentials, which are listed in **Supplementary Fig. S28**. Besides, all these results indicate that 2 mA cm^{-2} is larger than 1 mA cm^{-2} in the value of overpotential (**Supplementary Fig. S29**). Under the same current, the overpotential of the densified electrolyte is slightly greater than that of the conventional electrolyte when the cells are operating normally.

Revised manuscript: “However, the overpotential of the cell using the densified electrolyte is slightly larger than that of the cell using conventional electrolyte, which can be ascribed to the relatively lower ion conductivity of densified electrolyte. When raising the current density from 1 mA cm^{-2} to 2 mA cm^{-2} , the overpotential of the cell using densified electrolyte is increased appropriately, but the cycle stability is still guaranteed. The symmetric cell can still be stably cycled for more than 2000 h at 2 mA cm^{-2} in the densified electrolyte, whereas the overpotential increases sharply after only 200 hours in the conventional electrolyte (Fig. 5e).”

Fig. 5 | Theoretical calculation and electrochemical performance of Zn/Zn symmetric cells. **a** The free-energy of HER on Zn (101), Zn (100), Zn (002), and Pt (111). **b** The absorption energy of Zn atom at various zinc crystal planes. **c** The absorption energy of $Zn(H_2O)_6^{2+}$ at various zinc crystal planes. Zn/Zn symmetric cells operated at different conditions: **d** 1 mA cm^{-2} , 0.5 mAh cm^{-2} and **e** 2 mA cm^{-2} , 1 mAh cm^{-2} . **f** The XRD patterns of the zinc foil after 50 hours cycle. **g** The Coulombic efficiency of Zn/Ti half cells. **h** The corresponding charge-discharge curves at different cycles.

Repeated experiments:

Supplementary Fig. S28 | The electrochemical performance of Zn/Zn symmetric cells

at different current densities of: a) 1 mA cm^{-2} , 0.5 mAh cm^{-2} and b) 2 mA cm^{-2} , 1 mAh cm^{-2} .

Supplementary Fig. S29 | The comparison of Zn/Zn symmetric cells using densified electrolytes at different current densities.

9) The Figure S3 on DFT calculations must specify well the colors and the species absorbed, like H₂O in S3.

Answer: Thank you for your kind suggestion. We apologize that we have neglected this detail that the volume of H₂O is very small and the O element is the same color as the O element of SrTiO₃, which makes it difficult to distinguish. We have marked the sites of adsorbed H₂O in **Supplementary Fig. S5** (the original version of Supplementary Figure S3) for easy observation. Besides, we marked the sites of H adsorption in **Supplementary Fig. S10**.

Revised manuscript:

Supplementary Fig. S5 | The Top view of geometrical configurations of H₂O adsorbed on the various SrTiO₃ (110) plane.

Supplementary Fig. S10 | The geometrical configurations of the H adsorption energy.

10) There are some typos throughout the entire text. Such as, μAhcm^{-2} (pag 8, twice); Fig 4i and 4j (pag 9); etc.

Answer: Thank you for your very kind suggestion. We have corrected the typos including serial numbers, units, words, etc. throughout the manuscript as you suggested.

Revised manuscript:

“Inorganic compounds, such as metal oxides and inorganic salts, are mainly used to regulate the electrochemical behaviors at the interface between zinc anodes and electrolytes.”

“The evolution of zinc deposition in 2 M ZnSO₄ is shown in Fig. 4a.”

“When the deposition capacity reaches 500 μAh cm⁻², in addition to the obvious dendrites, large accumulations of by-products can also be observed on the surface of zinc foil.”

“In sharp contrast, the zinc foil deposited in the densified electrolyte exhibits a flat deposition behavior, as shown in Fig. 4b.”

“The deposited zinc formed a smoother and tighter surface without any by-products as the deposition capacity reaches 500 μAh cm⁻².”

“As shown in Fig. 4c, the heterogeneous deposition of zinc ions due to the non-uniformity of the electric field and the concentration field leads to the growth of zinc dendrites in conventional electrolyte.”

“These terrible situations take a dramatic turn in densified electrolytes, as shown in Fig. 4d.”

Reviewer #2:

In this manuscript, the authors present an innovative aqueous electrolyte design for zinc ion batteries. By incorporating SrTiO₃ particles into the conventional electrolyte, the authors were able to expand the thermodynamic stability window of the aqueous electrolytes and achieve non-dendritic Zn deposition with a preferred orientation of (002). Extensive experiments have been carried out to support the design. The results could be of general interest to the electrochemistry and battery community. While finding the results interesting, I still think certain analysis/interpretation remain to be revisited and revised with extra care. I would recommend publication of the manuscript if the authors could fully address my concerns. Below, I have detailed my questions and some minor comments,

Answer: Dear Reviewer #2, thank you very much for evaluating our manuscript and appreciating our work. We have explained all the questions and revised the manuscript per your suggestions.

1. The choice of SrTiO₃ as the additive seems to be an interesting one. Why SrTiO₃ was chosen in this study? What would be the design/screening principle for such inorganic additives? This is important to enlighten future study in a systemic way.

Answer: Thank you for your farsighted suggestion. There are two main considerations in our choice of SrTiO₃ as an additive. On the one hand, we found that Joachim Maier put forward the concept of “heterogeneous doping” in the organic liquid electrolytes. This concept involves adding one phase to another phase, mainly including solid-solid and solid-liquid, which can deliver the advantages of high ionic conductivity, solid-like appearance, and high transference number. And it has been applied in the lithium-ion batteries, like adding SiO₂, TiO₂, and Al₂O₃ into such organic electrolytes (e.g., LiPF₆ in EC/DMC). Therefore, we have chosen a lot of inorganic materials as additives to studies their performance on aqueous zinc ion batteries. On the other hand, we found that there are many inorganic materials, such as SiO₂, TiO₂, CaCO₃, BaTiO₃, were used as coating layers in the modification of the zinc anodes. However, over the long-term

repeated cycling of the batteries, the artificial coating layers are likely to be damaged or destroyed during Zn stripping/plating. Combined the concept of “heterogeneous doping” and those inorganic materials for coating layers, we have investigated and screened the properties of a variety of inorganic oxide materials, one of which is SrTiO₃. As for the design/screening principle for such inorganic additives, we mainly consider the acidity and basicity of the oxides. The family of inorganic oxide materials includes acidic oxides such as SiO₂, basic oxides such as TiO₂, SrTiO₃, and amphoteric oxides such as Al₂O₃. Oxides with different acidity and basicity have different properties. First, when added oxides particles to the conventional electrolytes, they have different effects with water molecules, anions and cations. Besides, they may affect the pH of the electrolytes, which has a significant impact on the electrochemical behavior of aqueous zinc ion batteries. In fact, we have systematically studied the electrochemical performance of various oxides in aqueous zinc-ion batteries. **Compared to other oxides, SrTiO₃ can significantly improve the specific capacity of the batteries and specially induce Zn deposition along the (002) crystal surface, thus deliver more outstanding performance of high-voltage aqueous Zn-ion batteries.**

2. For the name “densified electrolyte”, I am not sure “densified” is an appropriate term to describe the new electrolyte, rather it is misleading to some degree. I kept thinking which part of the electrolyte gets densified. (But in respect to the authors’ will and for the sake of simplicity, I will keep use “densified” in the following comments.)

Answer: Thank you for your kind suggestion. In fact, we aim to use the “densified electrolyte” to represent this new type of aqueous electrolytes composed conventional aqueous electrolyte and addition of oxides. Both aqueous solution and oxide can be various. Different types of oxides and particle parameters, as well as different oxide contents, can lead to different electrochemical performance of the produced electrolytes. But the common point is that with the addition of oxides, the density of the electrolyte increases (see **Fig.1d**). Intuitively, we used “densified electrolyte” in our concept studies.

In this manuscript, we think that “densified” has three meanings. First, the modified electrolyte becomes densified with higher density, higher viscosity, and a solid-like appearance. Second, the addition of oxides densified the water molecules among the oxide particles network due to the force between the oxide particles and water molecules, and weaken the force between water molecules, reducing the activity of water molecules. Third, the densified electrolyte optimizes zinc deposition on the surface of Zn anodes, resulting in a densified deposition layer during cycles.

Revised manuscript:

“It is worth mentioning that densified electrolytes can be composed of different aqueous electrolytes and various oxides, generally referring to the electrolyte with increased density after the addition of oxides (Fig. 1d).”

Fig. 1 | The properties of aqueous densified electrolyte. a The crystal structure of SrTiO₃. **b** The XRD spectra of the SrTiO₃ powder and its standard PDF card. **c** Schematic diagram of densified aqueous electrolytes. **d** Schematic illustration of densified electrolytes formed by increasing density of solution after the addition of oxide. **e** Comparison of Raman spectra of different electrolytes. Raman fit peaks of **f** ZnSO₄ electrolyte and **g** aqueous densified electrolyte. **h** The ratio of fitting strong H-bond area of electrolytes with various SrTiO₃ contents. **i** The Top view of geometrical configurations of H₂O adsorbed on the Ti atom of SrTiO₃ (110) plane. The boxes of molecular dynamics simulations with main solvated structure of Zn²⁺ in **j** ZnSO₄

electrolyte and k aqueous densified electrolyte.

3. In the Zn-Zn symmetric cell test, the overpotential in densified electrolyte becomes larger over cycling. The increasing overpotential is common in Li metal symmetric cell tests due to the build-up of residue SEI and depletion of electrolyte. But there should be no SEI on Zn anode as the authors have indicated. What could be the cause of this overpotential growth in Zn-Zn symmetric cells for densified electrolyte? From my perspective, an opposite trend in the overpotential (keep decreasing or at least remain constant) seems more reasonable, as Zn anode would have increased surface area upon repetitive deposition and dissolution.

Answer: Thank you for your kind suggestion. We apologize for not explaining the increasing overpotential in densified electrolyte. In the aqueous zinc ion battery, SEI does not exist in the conventional electrolyte, and the SrTiO₃ is almost insoluble in the conventional electrolyte, so it will not react with zinc metal anode to produce SEI, which can be proved by the EIS shown in Supplementary Fig. S25.

The overpotential increase of symmetric cells with densified electrolyte may be caused by settlement of SrTiO₃ particles on the outer surface of Zn metal, when cells have gone through a long calendar life. As shown in Supplementary Fig. S8, after soaking in densified electrolyte for 15 days, there is no by-product formation on the surface of zinc foil but there is a thin white layer. According to XRD results, this film is SrTiO₃ (Fig. 3f). Besides, the XRD patterns of the anodes obtained from Zn/MnO₂ cells after 500 cycles further proves this view (Fig. 6h).

Revised manuscript: “The drastically increasing overpotential can be attributed to the deterioration of the interface due to large accumulation of by-products produced by side reactions in the conventional electrolyte. In contrast, a slight increase in overpotential is also found in symmetric cells with densified electrolytes, this is caused by settlement of SrTiO₃ particles on the surface of zinc foils during a long-time cycle.”

Supplementary Fig. S25 | The EIS of full cells using various electrolytes: **a** before cycle and **b** after first cycle.

4. Closely coupled with my question 3, the stripping curve of ZnSO₄-SrTiO₃-500th in Fig. 5h indicates a similar polarized behavior. What could be the origin? Is Zn anode surface still “SEI” free in this new electrolyte?

Answer: Thank you for your kind suggestion. According to the answer to question 3, we believe that the increased overpotential of Zn/Ti half cells is also caused by partial settlement of SrTiO₃ on the outer surface of Zn metal. And we have made corresponding explanations in the manuscript.

Revised manuscript: “On the contrary, Fig. 5h shows that the efficiency of the cell using densified electrolyte is 81% in the first cycle, then gradually increases to 99.6% in the 500th cycle, and the high CEs can be stably maintained to more than 1000 cycles. With the increase of the cycle number, the overpotential increases slightly, which may be due to the partial settlement of SrTiO₃ on the electrodes due to the long-time operation.”

5. Fig. 5d and 5e, two different current densities and deposition capacities were investigated. I failed to find consistency in electrolyte ionic conductivity here. The overpotential of densified electrolyte at 2 mA/cm⁻² is lower than conventional electrolyte, while the overpotential at 1 mA/cm⁻² is higher than conventional electrolyte. Given that no SEI on Zn anode, and from Fig. 3, interfacial charge transfer resistance

in densified electrolyte is lower than that of conventional electrolyte, these results suggest that the electrolyte ionic conductivity has non-trivial variation. Could the authors measure the ionic conductivity of both electrolytes and compare?

Answer: We would like to apologize for that we accidentally reversed the data when we were processing and drawing graphic. We negligently reversed the data of densified electrolytes in Fig. 5d and 5e, that is, we put the data of 1 mA cm^{-2} of densified electrolyte in Fig. 5e (represent 2 mA cm^{-2}), but put the data of 2 mA cm^{-2} in Fig. 5d, which led to the inconsistency of the results. The correct data has been updated in Fig. 5. Meanwhile, to ensure the reliability of the data, we repeated several new experiments to compare the overpotentials, which are listed in the **Supplementary Fig. S28** below in this revision. Besides, all these results indicate that 2 mA cm^{-2} is larger than 1 mA cm^{-2} in the value of overpotential (**Supplementary Fig. S29**). Under the same current, the overpotential of the densified electrolyte is slightly greater than that of the conventional electrolyte when the cells are operating normally.

In addition, we tested the ion conductivity of various electrolytes with an ionic conductivity instrument as you suggested (**Supplementary Fig. S3**). It can be found that the conductivity of densified electrolytes decreases with the increase of SrTiO_3 content.

Revised manuscript: “However, the overpotential of the cell using the densified electrolyte is slightly larger than that of the cell using conventional electrolyte, which can be ascribed to the relatively lower ion conductivity of densified electrolyte. When raising the current density from 1 mA cm^{-2} to 2 mA cm^{-2} , the overpotential of the cell using densified electrolyte is increased appropriately, but the cycle stability is still guaranteed. The symmetric cell can still be stably cycled for more than 2000 h at 2 mA cm^{-2} in the densified electrolyte, whereas the overpotential increases sharply after only 200 hours in the conventional electrolyte (Fig. 5e).”

Fig. 5 | Theoretical calculation and electrochemical performance of Zn/Zn symmetric cells. a The free-energy of HER on Zn (101), Zn (100), Zn (002), and Pt (111). **b** The absorption energy of Zn atom at various zinc crystal planes. **c** The absorption energy of $Zn(H_2O)_6^{2+}$ at various zinc crystal planes. Zn/Zn symmetric cells operated at different conditions: **d** 1 mA cm^{-2} , 0.5 mAh cm^{-2} and **e** 2 mA cm^{-2} , 1 mAh cm^{-2} . **f** The XRD patterns of the zinc foil after 50 hours cycle. **g** The Coulombic efficiency of Zn/Ti half cells. **h** The corresponding charge-discharge curves at different cycles.

repeated experiments:

Supplementary Fig. S28 | The electrochemical performance of Zn/Zn symmetric cells

at different current densities of: a) 1 mA cm⁻², 0.5 mAh cm⁻² and b) 2 mA cm⁻², 1 mAh cm⁻².

Supplementary Fig. S29 | The comparison of Zn/Zn symmetric cells using densified electrolytes at different current densities.

Supplementary Fig. S3 | **a** The conductivities of electrolytes with different SrTiO₃ contents. **b** their corresponding viscosities tested on a rotational rheometer at a shear rate of 1000 s⁻¹.

6. Was the same amount of liquid portion of the electrolyte used or the same total weight of electrolyte used in all these measurements?

Answer: We used the same volume of 400 μL electrolyte in symmetric cells, half cells and full cells, which were sucked out using a pipette. We have added the detail to the methods section.

Revised manuscript: “All of the coin cells were assembled with 400 μL electrolyte

and studied on battery testing instruments (Land, China).”

7. In Zn/MnO₂ full cell with conventional electrolyte (2M ZnSO₄, 0.1M MnSO₄), the redox reaction is a single-electron redox reaction (Mn(IV) - Mn(III)). The theoretical specific capacity of MnO₂ in single-electron redox reaction is 308 mAh/g. However, in densified electrolyte, the reported specific capacities apparently exceed this value, seems to suggest the coexistence of both single-electron redox and two-electron redox (Mn(IV) - Mn(II)) reactions. The two-electron pathway corresponding to MnO₂-Mn²⁺(aq) conversion only happens at quite acidic condition and does not seem possible in the densified electrolyte. I suspect that part of the Mn²⁺ from the electrolyte (0.1M MnSO₄) got oxidized into MnO₂ since in densified electrolyte the cell was charged up to 2 V, therefore effectively increase the actual MnO₂ loading. I would recommend the authors to check cycling behavior in electrolytes without 0.1M MnSO₄. If such high specific capacity still exists, the authors might need to explain the mechanism of the potential two-electron redox in the densified electrolyte.

Answer: Thank you very much for your useful suggestions. Combined with your comments and some recently published articles (10.1002/adma.202300053, 10.1039/D1EE03547A, 10.1016/j.ensm.2019.12.021), we have done a large number of experiments and found that densified electrolyte provides an additional electrochemical reaction ($x\text{Zn}^{2+} + y\text{Mn}^{2+} + \text{H}_2\text{O} \rightleftharpoons \text{Zn}_x\text{Mn}_y\text{O} + 2\text{H}^+ + (2-2x-2y)\text{e}^-$).

Firstly, according to your suggestion, we studied the electrochemical performance of Zn/MnO₂ full cells without 0.1M MnSO₄. It is found that the electrochemical performances of the cells are significantly decreased, which indicates that MnSO₄ plays an indispensable role in both electrolytes (**Supplementary Fig. S16a**). To verify whether the Mn²⁺ in the densified electrolyte is oxidized into MnO₂, we studied the performance of Zn/MnO₂ full cells without ZnSO₄, in which only the insertion of H⁺ into MnO₂ is involved. As shown in **Supplementary Fig. S16b**, the performance of the cell using densified electrolyte is even worse than conventional electrolyte. This suggests two things: First, the hypothesis that the Mn²⁺ in the densified electrolyte is oxidized into MnO₂ may not be correct; Second, the H⁺ content of the densified

electrolyte is lower than that of the conventional electrolyte, that is, the pH of the densified electrolyte may be higher than that of conventional electrolyte.

Next, we tested the pH of conventional electrolyte and densified electrolytes with various mass ratios of SrTiO₃. **Supplementary Fig. S15** demonstrates that once SrTiO₃ is added, the pH values of the densified electrolytes increase rapidly to more than 5.5, and the pH of the densified electrolyte with 50 wt.% increases from 3.4 to 5.76. This may be due to the fact that SrTiO₃ is an alkaline oxide, which inhibits hydrolysis of the densified electrolytes and thus increases the pH values.

Then, we hypothesized that the exceeded capacity may be due to additional electrochemical reaction that occurred after the pH increase. According to some published article (10.1002/adma.202300053, 10.1039/D1EE03547A, 10.1016/j.ensm.2019.12.021), when pH is around 6, in addition to the co-insertion of H⁺/Zn²⁺, the reaction ($x\text{Zn}^{2+} + y\text{Mn}^{2+} + \text{H}_2\text{O} \rightleftharpoons \text{Zn}_x\text{Mn}_y\text{O} + 2\text{H}^+ + (2-2x-2y)\text{e}^-$) also occurs. We designed an experiment to test and verify this hypothesis. The cathode composed of Super P and PVDF without any MnO₂ was used, and the produced cell was defined as SP cell. **Supplementary Fig. S17a** shows the performance of the SP cell using conventional electrolyte (2M ZnSO₄, 0.1M MnSO₄). When the charging cutoff voltage is 1.8 V, the cell has almost no capacity. When the voltage is raised to 2 V, the specific capacity is about 15 mAh g⁻¹ (we set the active material mass to 1 mg). The SP cell using densified electrolyte exhibits a low specific capacity of 20 mAh g⁻¹ under the charge cut-off voltage of 1.8 V, but deliver a higher specific capacity about 100 mAh g⁻¹ under voltage of 2.0 V. (**Supplementary Fig. S17b**). This indicates that increasing the voltage to 2.0 V is an essential condition for obtaining high capacity. More importantly, we found that there is significantly difference in the charge-discharge curves. The charge-discharge curves of the first two circles of the SP cell using conventional electrolyte have distinct plateaus around 1.99V (**Supplementary Fig. S17c**), which is caused by the oxidation of small amounts of Mn²⁺ to MnO₂ (10.1002/anie.201904174). In contrast, the cell with densified electrolyte provides a longer platform below 1.7 V, and the curve is slowly raised to 2.0 V, indicating that there is almost no electrochemical behavior of Mn²⁺ oxidation to MnO₂ at high voltage

plateaus. This is consistent with the charge-discharge curves in the whole Zn/MnO₂ full cells (**Supplementary Fig. S18**), which indicates that the electrochemical behavior in conventional electrolyte is indeed different from that in densified electrolyte.

To further analyze the electrochemical behaviors, the cyclic voltammetry tests of SP cells were carried out at a scan rate of 0.1 mV s⁻¹ (**Supplementary Fig. S19**). Since the cathode contains only SP without any MnO₂, the first discharge curve does not have any current. During the first charge, the SP cell using conventional electrolyte has a sharp peak at around 2.0 V, which leads to the generation of MnO₂. In the second discharge curve, the SP cell with conventional electrolyte shows a similar peak to that of the Zn/MnO₂ full cells, that is, a smaller peak around 1.4 V and a larger peak around 1.2 V. In the second charge curve, the SP cell using conventional electrolyte still shows a similar peak to that of the Zn/MnO₂ full cells, and with a sharp peak at 2.0 V to produce MnO₂. This is further evidenced by the third cycle of the discharge curve in the conventional electrolyte, where the peak currents at both 1.4 V and 1.2 V increase, indicating more MnO₂ will be produced once charging to 1.99 V (**Supplementary Fig. S19a**). As a contrast, there is only a small peak at 2.0 V, indicating trace amounts of MnO₂ could be produced at 2.0 V in the first charge, while a special peak at 1.6 V in densified electrolyte is also visible, corresponding to Zn_xMn_yO production (**Supplementary Fig. S19b**). The 1.4 V peak is significantly larger than the 1.2 V peak in the next discharge curve (**Supplementary Fig. S19b**), which is consistent with the CV curves of Zn/MnO₂ full cells (**Fig. 6e**), indicating that Zn_xMn_yO does generate in the previous process. More importantly in the second cycle in **Supplementary Fig. S19b**, the SP cell using densified electrolyte only shows one peak at 1.65V without any peak close to 2.0 V, suggesting that only a small amount of MnO₂ is produced in the first charge, and there would be no any MnO₂ produced after the second cycle. Furthermore, the densified electrolyte remains almost unchanged during the third charge curve (**Supplementary Fig. S19b**), but the conventional electrolyte shows an additional peak at 1.65 V (**Supplementary Fig. S19a**), which may be due to the increased pH of the conventional electrolyte leading to Zn_xMn_yO production (10.1038/s41467-022-29987-x, 10.1002/sml.202005406).

Finally, we conducted XRD tests on the cathode of the SP cells charged to 2.0 V for the first time. In the conventional electrolyte, we only detected the presence of the phase of the stainless steel, and did not detect MnO₂, which may be because the amount is too small to be detected (**Supplementary Fig. S20a and S20b**). In contrast, in addition to the peak for stainless steel and SrTiO₃ in the densified electrolyte, we also found some additional peaks between 26 and 30 degrees, which are attributed to Zn_xMn_yO (**Supplementary Fig. S20c and S20d**).

In summary, all of these results demonstrate the existence of an electrochemical reaction in densified electrolytes, that is: $x\text{Zn}^{2+} + y\text{Mn}^{2+} + \text{H}_2\text{O} \rightleftharpoons \text{Zn}_x\text{Mn}_y\text{O} + 2\text{H}^+ + (2-2x-2y)\text{e}^-$. And we have revised the manuscript according to your suggestions.

Revised manuscript: “It is worth mentioning that the specific capacity of Zn/MnO₂ full cells with densified electrolyte has exceeded the theoretical value of 308 mAh g⁻¹. This is because the electrolyte pH increased from 3.4 to 5.76 after the addition of SrTiO₃ (Supplementary Fig. S15), resulting in additional electrochemical reaction ($x\text{Zn}^{2+} + y\text{Mn}^{2+} + \text{H}_2\text{O} \rightleftharpoons \text{Zn}_x\text{Mn}_y\text{O} + 2\text{H}^+ + (2-2x-2y)\text{e}^-$).^{43, 44, 45} To exclude the interference of the active material MnO₂, the cathode containing only Super P and PVDF is designed to assemble the cell (note as SP cell). It is found that SP cells can only make additional capacity contributions when the densified electrolytes contain MnSO₄, ZnSO₄, SrTiO₃ and is charged to 2.0 V (Supplementary Fig. S16 and S17). The charge-discharge curves of the SP cell using conventional electrolyte have distinct plateaus around 1.99 V, which is caused by the oxidation of small amounts of Mn²⁺ to MnO₂.⁴⁶ In contrast, the cells with densified electrolytes have a longer platform below 1.7 V, and the curve is slowly raised to 2.0 V. This is consistent with the charge-discharge curves in the whole Zn/MnO₂ full cells (Supplementary Fig. S18), which indicates that the electrochemical behavior of conventional electrolyte is indeed different from that of densified electrolyte. The cyclic voltammetry tests of the SP cells were carried out to further study the electrochemical behavior. As shown in Supplementary Fig. S19a, after the first charge to 2.0 V, the CV curves of SP cell with conventional electrolyte is almost the same as that of Zn/MnO₂ full cells, indicating that Mn²⁺ is oxidized to MnO₂. Peculiarly,

a weak peak corresponding to Zn_xMn_yO can be observed at the third cycle of the CV curve, which is because the pH increases due to side reactions, further indicating that the Zn_xMn_yO would be produced in electrolytes with higher pH^{47, 48}. In contrast, the densified electrolyte showed no sharp peaks at 2.0 V indicating almost no MnO_2 production, but distinct peaks at 1.65 V and 1.35 V, which correspond to the redox reaction of Zn_xMn_yO (Supplementary Fig. S19b). The XRD patterns of cathodes of SP cells after charging to 2.0 V provides further evidence (Supplementary Fig. S20). In the conventional electrolyte, the XRD pattern only shows that the presence of the stainless steel but without MnO_2 , which may be due to the amount is too small to be detected. In contrast, in addition to the peak for stainless steel and $SrTiO_3$ in the densified electrolyte, the XRD pattern exhibits several peaks, which are attributed to Zn_xMn_yO . Overall, all the results show that the pH can be stabilized at about 5.8 after the addition of $SrTiO_3$, which leads to a continuous and reversible reaction: $xZn^{2+} + yMn^{2+} + H_2O \rightleftharpoons Zn_xMn_yO + 2H^+ + (2-2x-2y)e^-$.”

“43. Yang H, *et al.* Protocol in Evaluating Capacity of Zn–Mn Aqueous Batteries: A Clue of pH. *Adv. Mater* DOI: 10.1002/adma.202300053 (2023).

44. Yang H, *et al.* The origin of capacity fluctuation and rescue of dead Mn-based Zn–ion batteries: a Mn-based competitive capacity evolution protocol. *Energy Environ. Sci.* **15**, 1106-1118 (2022).

45. Vaiyapuri S, *et al.* The dominant role of Mn^{2+} additive on the electrochemical reaction in $ZnMn_2O_4$ cathode for aqueous zinc-ion batteries. *Energy Stor. Mater.* **28**, 407-417 (2019).

46. Chao D, *et al.* An Electrolytic Zn- MnO_2 Battery for High-Voltage and Scalable Energy Storage. *Angew. Chem., Int. Ed.* **58**, 7823-7828 (2019).

47. Yangmoon K, *et al.* Corrosion as the origin of limited lifetime of vanadium oxide-based aqueous zinc ion batteries, *Nat. Commun.* **13**, 2371 (2022).

48. Sung J K, *et al.* Unraveling the Dissolution-Mediated Reaction Mechanism of α - MnO_2 Cathodes for Aqueous Zn-Ion Batteries, *Small.* **16**, 2005406 (2020).”

Supplementary Fig. S15 | The pH of conventional electrolyte and densified electrolytes with different SrTiO₃ contents.

Supplementary Fig. S16 | The galvanostatic cycling performance of Zn/MnO₂ cells at a current density of 1 A g⁻¹ in various electrolytes: **a** without MnSO₄ and **b** without ZnSO₄.

Supplementary Fig. S17 | The galvanostatic cycling performance of Zn/SP cells (set the active material mass is 1 mg) under different charge cut-off voltage at a current density of 1 A g^{-1} in **a** conventional electrolytes and **b** densified electrolyte; The first two charge-discharge curves of SP cells under charge cut-off voltage of 2.0 V in **c** conventional electrolytes and **d** densified electrolytes.

Supplementary Fig. S18 | The first two charge and discharge curves of Zn/MnO₂ full cells in **a** conventional electrolyte and **b** densified electrolyte.

Supplementary Fig. S19 | The cyclic voltammetry (CV) profiles of SP cells at a scan rate of 0.1 mV s^{-1} : **a** in the conventional electrolyte and **b** in the densified electrolyte.

Supplementary Fig. S20 | The investigation on the redox behavior of the first charge to 2.0 V: **a** the XRD pattern and **b** the corresponding local magnification ranges (marked area) of the SP cathode charged in the conventional electrolyte; **c** the XRD pattern and **d** the corresponding local magnification ranges (marked area) of the SP cathode charged in the densified electrolyte.

Minor comments:

1. In Fig. 5g, the two electrolytes are mislabeled.

Answer: Thank you for your very kind suggestion. We have corrected **Fig. 5g** in the manuscript as you suggested.

Revised manuscript:

Fig. 5 | Theoretical calculation and electrochemical performance of Zn/Zn symmetric cells. a The free-energy of HER on Zn (101), Zn (100), Zn (002), and Pt (111). **b** The absorption energy of Zn atom at various zinc crystal planes. **c** The absorption energy of $Zn(H_2O)_6^{2+}$ at various zinc crystal planes. Zn/Zn symmetric cells operated at different conditions: **d** 1 mA cm^{-2} , 0.5 mAh cm^{-2} and **e** 2 mA cm^{-2} , 1 mAh cm^{-2} . **f** The XRD patterns of the zinc foil after 50 hours cycle. **g** The Coulombic efficiency of Zn/Ti half cells. **h** The corresponding charge-discharge curves at different cycles.

2. When the authors reported specific capacity, I assume that it is calculated based on MnO₂ cathode, but please clarify the specific capacity of what in the manuscript.

Answer: Thank you for your kind suggestion. The specific capacity calculation was indeed based on the mass of the active material MnO₂. And according to your suggestion, we have clarified this point in the method section.

Revised manuscript: “Cathodes were prepared by mixing PVDF, super P, and MnO₂ in a mass ratio of 1:2:7, and stirring for 12 hours after adding NMP. The produced slurry was coated on stainless steel with active materials (MnO₂) loading of 0.8-1 mg cm⁻² or 4.0 mg cm⁻². Then the stainless steel coated with the active material was dried at 80 °C overnight in a vacuum oven and was punched into small round pieces of 1 cm in diameter. The specific capacity was calculated based on the mass of the active material MnO₂.”

3. Page 4, line 114, the authors indicate that the densified electrolyte has “good mechanical strength”. Is this property relevant to Zn metal anode or full cell performance? I did not find either measurement on this property or clear logical relation in the later discussion with the battery performance.

Answer: Thank you for your kind suggestion. Here, we want to use “good mechanical strength” to describe that densified electrolytes have some solid-like characteristics. As shown in Supplementary Fig. S2, densified electrolytes have semi-solid physical properties compared to conventional electrolytes. But to be more rigorous, we have made appropriate corrections according to your suggestion in the revised manuscript. For example, we moved the relative discussion into the conclusion part to demonstrate the potential advantage of this kind of densified electrolyte on the mechanical strength for future studies. In our mind, further screen on the types of oxides and their particle parameters would cause the enhanced mechanical strength or even lead to separator-free zinc batteries, that is, similar to an all-solid electrolyte.

Revised manuscript:

“The prepared aqueous densified electrolyte significantly improves the electrochemical performance of high-voltage zinc ion batteries, providing a new design concept and

solution for the electrolyte optimization of aqueous rechargeable batteries. More importantly, further screen on the types of oxides and their particle parameters would enhance solid-like characteristic with potentially good mechanical strength and even lead to separator-free zinc batteries.”

Supplementary Fig. S2 | The optical photos of the dense electrolyte.

REVIEWER COMMENTS

Reviewer #1 (Remarks to the Author):

In this version, the authors have reviewed the article completely. I believe that the modifications carried out perfectly answer my doubts and that the article has been greatly improved following the suggestions and comments that the reviewers indicated. Consequently, I think that this work is prepared to be accepted by the journal Nature Communications.

Reviewer #2 (Remarks to the Author):

I appreciate the authors' efforts, and great improvements achieved in the revision. The manuscript is in a better place in terms of readiness for publication. However, I still find one important question not fully addressed – Comment No.7 from my 1st round review. First, I appreciated the authors' effort and detailed experiments on addressing this issue. However, the experiments and literature reviews they did supported what I pointed out – the oxidation (or redox) of Mn²⁺ in the electrolyte contributed to the increased capacity. According to the authors, there existed one reaction below that helped explain the extra capacity,

$$x\text{Zn}^{2+} + y\text{Mn}^{2+} + \text{H}_2\text{O} \rightleftharpoons \text{Zn}_x\text{Mn}_y\text{O} + 2\text{H}^+ + (2-2x-2y)e^-$$

But there is no Zn_xMn_yO to begin with, the only Mn²⁺ available was in the electrolyte. This actually corresponded to the redox reactions that gave the extra capacity I was talking about. Admittedly, in these experiments, no obvious capacity was observed for conventional electrolytes even at 2V, but large capacities or oxidation peaks in CV were observed in densified electrolytes. This is because the pH of the electrolyte increased in the densified electrolytes as the authors have measured. According to the Pourbaix diagram for Mn, the oxidation reaction voltage for Mn(II)-Mn(IV) is lower in more basic conditions. Therefore, in the given voltage range, the Mn²⁺ from MnSO₄ got oxidized in the densified electrolyte to give extra capacity over the nominal theoretical limit. While the authors could claim that the increase was the result of using densified electrolytes, but the origin was in the electrolyte salt additives that got oxidized. I think it is necessary to clarify this point in the manuscript, otherwise the comparison of specific capacities can be misleading in the way that people might falsely attribute the extra capacity to the original solid MnO₂ loading.

Point-by-point response

Reviewer #1:

In this version, the authors have reviewed the article completely. I believe that the modifications carried out perfectly answer my doubts and that the article has been greatly improved following the suggestions and comments that the reviewers indicated. Consequently, I think that this work is prepared to be accepted by the journal Nature Communications.

Answer: Dear Reviewer #1, Thank you very much for your affirmation of our revision, and thank you very much for your valuable suggestions that make our manuscript improved.

Reviewer #2:

I appreciate the authors' efforts, and great improvements achieved in the revision. The manuscript is in a better place in terms of readiness for publication. However, I still find one important question not fully addressed - Comment No.7 from my 1st round review. First, I appreciated the authors' effort and detailed experiments on addressing this issue. However, the experiments and literature reviews they did supported what I pointed out - the oxidation (or redox) of Mn^{2+} in the electrolyte contributed to the increased capacity. According to the authors, there existed one reaction below that helped explain the extra capacity,

But there is no Zn_xMn_yO to begin with, the only Mn^{2+} available was in the electrolyte. This actually corresponded to the redox reactions that gave the extra capacity I was talking about.

Admittedly, in these experiments, no obvious capacity was observed for conventional electrolytes even at 2V, but large capacities or oxidation peaks in CV were observed in densified electrolytes. This is because the pH of the electrolyte increased in the densified electrolytes as the authors have measured. According to the Pourbaix diagram for Mn, the oxidation reaction voltage for Mn(II)-Mn(IV) is lower in more basic

conditions. Therefore, in the given voltage range, the Mn^{2+} from MnSO_4 got oxidized in the densified electrolyte to give extra capacity over the nominal theoretical limit.

While the authors could claim that the increase was the result of using densified electrolytes, but the origin was in the electrolyte salt additives that got oxidized. I think it is necessary to clarify this point in the manuscript, otherwise the comparison of specific capacities can be misleading in the way that people might falsely attribute the extra capacity to the original solid MnO_2 loading.

Answer: Dear Reviewer #2, thank you very much for appreciating our revision, and thank you for your comments to help us improve the manuscript. As you say, it is necessary to state that the extra capacity over the theoretical value arises from the oxidation of Mn^{2+} in the electrolyte, otherwise it may indeed mislead the reader that all of the capacity comes from the original solid MnO_2 loading. We agree with you and have clarified this point in appropriate places in the manuscript.

Revised manuscript:

“It is worth mentioning that the specific capacity of Zn/MnO_2 full cells with densified electrolyte has exceeded the theoretical value of 308 mAh g^{-1} , which is because the Mn^{2+} from MnSO_4 additive in the densified electrolyte is oxidized when the cell is charged to 2 V, thus contributing additional capacity beyond the original solid MnO_2 . The redox reaction of Mn^{2+} in the densified electrolyte originates from the pH increased from 3.4 to 5.76 after the addition of SrTiO_3 (Supplementary Fig. S15), resulting in additional electrochemical reaction ($x\text{Zn}^{2+} + y\text{Mn}^{2+} + \text{H}_2\text{O} \rightleftharpoons \text{Zn}_x\text{Mn}_y\text{O} + 2\text{H}^+ + (2-2x-2y)\text{e}^-$)”.

“Overall, these results show that the pH can be stabilized at about 5.8 after the addition of SrTiO_3 , which leads to a continuous and reversible reaction: $x\text{Zn}^{2+} + y\text{Mn}^{2+} + \text{H}_2\text{O} \rightleftharpoons \text{Zn}_x\text{Mn}_y\text{O} + 2\text{H}^+ + (2-2x-2y)\text{e}^-$, accompanied by additional capacity from the produced solids viz. $\text{Zn}_x\text{Mn}_y\text{O}$ ”.

REVIEWERS' COMMENTS

Reviewer #2 (Remarks to the Author):

The authors have addressed my comments well in this revision and I think that this work is ready for publication in Nature Communications.

Point-by-point response

Reviewer #2:

The authors have addressed my comments well in this revision and I think that this work is ready for publication in Nature Communications.

Answer: Dear Reviewer #2, Thank you so much for approving our revision, and thank you very much for your valuable suggestions, which significantly enhances the quality of our manuscript.